# Topological protection by local support symmetry and destructive interference

Jun-Won Rhim ®[1] ✉, Jaeuk Seo ®[2], Seongjun Mo[3], Hoonkyung Lee ®[3], Sejoong Kim[4] & B. Andrei Bernevig ®[5,6,7] ✉

Conventionally, symmetry-protected topological phases and band crossings are protected by global symmetries acting on the entire system. Here, we show that symmetries preserved only on a partial region of a system, termed local support symmetries, can protect topological features of the full system, even in the presence of symmetry-breaking couplings. We establish a unified framework by deriving explicit conditions for such protection in both insulating and metallic phases and show that destructive interference of Bloch wave functions plays a key role. Using representative tight-binding models, we demonstrate band crossings and topological bands protected by local support crystalline and time-reversal symmetries, and further present a realistic material realization in a fluorinated biphenylene network, where a band crossing is protected by a local support $C_2$ symmetry.

Triggered by the discovery of the topological insulator[1–8], researchers have almost exhaustively investigated a myriad of symmetry-protected topological (SPT) phases and topological semimetals[8–23]. Symmetry plays a critical role in the stabilization of those topological phases. For example, the nontrivial topology in $\mathbb{Z}_2$ topological insulators and topological crystalline insulators is protected by time-reversal and crystalline symmetries, respectively[1,4,8,22]. On the other hand, band-crossings, such as nodal points, nodal lines, and nodal planes, can circumvent gap opening with the aid of various crystalline symmetries, such as a mirror and various nonsymmorphic symmetries[24–44]. In each of these cases, the symmetry operations uniformly affect the entire system, as illustrated in Fig. 1.

In this paper, we demonstrate that topological protections can persist even when a system respects a symmetry in one part of it ($\mathcal{S}_1$) while the symmetry is broken in another ($\mathcal{S}_2$), as plotted schematically in Fig. 1. Such a symmetry is called a local support symmetry (LSS). For instance, we illustrate the existence of insulators lacking time-reversal symmetry (TRS) yet exhibiting occupied bands characterized by a well-defined $\mathbb{Z}_2$ invariant, protected by a local support time-reversal symmetry. We also exhibit semimetal examples hosting nodal points protected by local support $C_2$ or nonsymmorphic

symmetries. It is crucial to emphasize that the LSS alone cannot preserve these topological characteristics; rather, the hopping processes between two parts of the system ($\mathcal{S}_1$ and $\mathcal{S}_2$), denoted by $\mathbf{h}_{12}$, must satisfy a certain condition as follows. On the other hand, a band-crossing along an LSS-invariant momentum line can be protected if the column vectors of $\mathbf{h}_{12}$ are proportional to one of the eigenvectors of the unitary matrix representing the LSS. We note that, in many cases, these conditions for the inter-part coupling ($\mathbf{h}_{12}$) can be achieved through destructive interference of a Bloch wave function, a phenomenon commonly associated with stabilizing flat bands[45–53]. Consequently, the amplitude of the Bloch wave function relevant to the LSS protection becomes precisely zero at sites belonging to $\mathcal{S}_2$, allowing for the application of the conventional protection mechanism of SPT phases only to $\mathcal{S}_1$. This highlights the crucial role of destructive interference in this topological protection mechanism. Finally, we propose that Dirac points protected by the local support $C_2$ symmetry can be observed in the biphenylene network[54] fluorinated periodically. While the absorbed fluorine atoms break $C_2$ symmetry in the entire system, a part of the system respects $C_2$ symmetry. Moreover, the inter-part coupling satisfies the conditions for the destructive interference mentioned above, so that the

[1]Department of Physics, Ajou University, Suwon, Korea. [2]Department of Physics, Korea Advanced Institute of Science and Technology, Daejeon, Korea. [3]Department of Physics, Konkuk University, Seoul, Korea. [4]Department of Electronic and Electrical Convergence Engineering, Hongik University, Sejong, Republic of Korea. [5]Department of Physics, Princeton University, Princeton, NJ, USA. [6]Donostia International Physics Center, Donostia-San Sebastian, Spain. [7]IKERBASQUE, Basque Foundation for Science, Bilbao, Spain. ✉e-mail: jwrhim@ajou.ac.kr; bernevig@princeton.edu

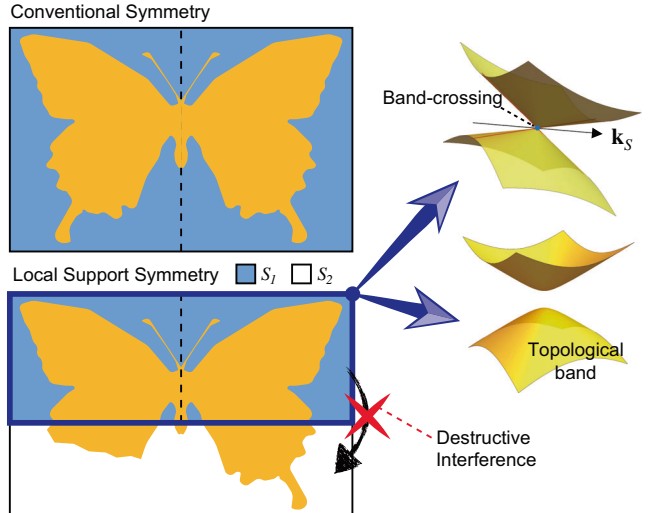

**Fig. 1 | Concept of local support symmetry and symmetry-protected band features.** Conventionally, SPTs and band-crossings are protected by a global symmetry that uniformly acts on the entire system. For example, the orange object preserves a $C_2$ symmetry about the dashed line throughout the whole region. In the case of the local support symmetry (LSS), the symmetry operation is applied to part $S_1$ (blue region) of the system while the other part $S_2$ (white region) is unaffected. With the aid of destructive interference, which can prevent the propagation of a Bloch wave function into $S_2$ from $S_1$, the LSS in $S_1$ can still protect band-crossings or band topology.

Dirac nodes can be protected by the local support $C_2$ symmetry even after the fluorination.

## Results

In this section, we demonstrate that a topological phase protected by symmetry can remain robust even when coupled to a system that breaks this symmetry, provided that the coupling meets a specific condition. We highlight that destructive interference can be crucial in satisfying this condition. We consider a system with $N$ orbitals per unit cell, which is partitioned into two distinct parts, labeled as $S_1$ and $S_2$, each comprising $N_1$ and $N_2$ orbitals, respectively. There is no overlap of orbitals between these two components, and $N_1 + N_2 = N$ (generalizations are possible for different graphs). Then, the Bloch Hamiltonian is given by

$$\mathbf{H}(\mathbf{k}) = \begin{pmatrix} \mathbf{h}_1(\mathbf{k}) & \mathbf{h}_{12}(\mathbf{k}) \\ \mathbf{h}_{21}(\mathbf{k}) & \mathbf{h}_2(\mathbf{k}) \end{pmatrix}, \tag{1}$$

where $\mathbf{h}_i(\mathbf{k})$ is a $N_i \times N_i$ sub-Hamiltonian matrix describing $S_i$-part and $\mathbf{h}_{12}(\mathbf{k}) = \mathbf{h}_{21}(\mathbf{k})^\dagger$ is a $N_1 \times N_2$ matrix representing the inter-part coupling. We assume that $S_1$ is in a topological phase protected by a symmetry, such as time-reversal and mirror symmetries. We represent the corresponding symmetry operation by $g$. On the other hand, $S_2$ does not respect the same symmetry. Namely, we consider an LSS for subsystem $S_1$. The symmetry representation corresponding to the LSS is denoted by $\mathbf{D}_\mathbf{k}(g)$. Depending on the type of the symmetry operation, $\mathbf{D}_\mathbf{k}(g)$ can be an $N_1 \times N_1$ unitary matrix or its multiplication with a complex conjugation $K$. This LSS representation satisfies $\mathbf{h}_1(\mathbf{k}) = \mathbf{D}_\mathbf{k}(g)\mathbf{h}_1(g^{-1}\mathbf{k})\mathbf{D}_\mathbf{k}(g^{-1})$ while the same symmetry is violated in the entire system described by the full Hamiltonian $\mathbf{H}(\mathbf{k})$. In the first two subsections below, we identify the conditions under which topological phases can be protected by LSSs, applicable across arbitrary spatial dimensions. We then apply the general theory to several example tight-binding models and a realistic system (DFT analysis).

## General theory: insulating case

We first consider band insulators protected by LSSs. We assume that the $N_1 \times N_1$ sub-Hamiltonian $\mathbf{h}_1(\mathbf{k})$ possesses a topological class relevant to the considered symmetry. The corresponding topological invariant, such as the $\mathbb{Z}_2$ index for a topological insulator and the mirror Chern number, can be calculated from the eigenvectors of the occupied bands of $\mathbf{h}_1(\mathbf{k})$, denoted by $\mathbf{v}_{1,n}^{(o)}(\mathbf{k})$. On the other hand, the eigenvectors of the unoccupied bands of $\mathbf{h}_1(\mathbf{k})$ are represented by $\mathbf{v}_{1,n}^{(u)}(\mathbf{k})$. Here, $n$ is the band index, and we assume that there are $N_1^{(o)}(N_1^{(u)})$ number of occupied(unoccupied) bands satisfying $N_1^{(o)} + N_1^{(u)} = N_1$. Since $\mathbf{v}_{1,n}^{(o)}(\mathbf{k})$ and $\mathbf{v}_{1,n}^{(u)}(\mathbf{k})$ correspond to different bands of the same Hamiltonian $\mathbf{h}_1(\mathbf{k})$, they satisfy $\mathbf{v}_{1,n}^{(u)\dagger}(\mathbf{k}) \cdot \mathbf{v}_{1,m}^{(o)}(\mathbf{k}) = 0$. As described above, we assume that the $N_2 \times N_2$ sub-Hamiltonian $\mathbf{h}_2(\mathbf{k})$ does not respect the protective symmetry of the $S_1$ part. Turning on the inter-part coupling $\mathbf{h}_{12}(\mathbf{k})$, $N_1^{(o)}$ number of eigenvectors among $N$ eigenvectors of the $N \times N$ full Hamiltonian $\mathbf{H}(\mathbf{k})$ can be of the form

$$\mathbf{V}_n^{(o)}(\mathbf{k}) = \begin{pmatrix} \mathbf{v}_{1,n}^{(o)}(\mathbf{k}) \\ \mathbf{O}_{N_2} \end{pmatrix}, \tag{2}$$

where $\mathbf{O}_{N_2}$ is a $N_2 \times 1$ zero matrix, iff the inter-part coupling satisfies

$$\mathbf{h}_{21}(\mathbf{k})\mathbf{v}_{1,n}^{(o)}(\mathbf{k}) = 0. \tag{3}$$

Here, $n$ runs from 1 to $N_1^{(o)}$ as in the uncoupled case. Note that the eigenenergy of $\mathbf{h}_1(\mathbf{k})$ corresponding to $\mathbf{v}_{1,n}^{(o)}(\mathbf{k})$ is precisely identical to that of $\mathbf{H}(\mathbf{k})$ corresponding to $\mathbf{V}_n^{(o)}(\mathbf{k})$. This implies that if the bands corresponding to $\mathbf{V}_n^{(o)}(\mathbf{k})$ are below the Fermi level and additional bands originating from $\mathbf{h}_2(\mathbf{k})$ do not affect the topological character of the gap (for example, by being above the gap), one obtains the same topological invariant for the entire system identical to the subsystem $\mathbf{h}_1(\mathbf{k})$ even though the protective symmetry is broken in the entire system due to its coupling to $\mathbf{h}_2(\mathbf{k})$. We describe this type of wave function in Eq. (2) as compactly supported because its amplitudes are strictly zero in $S_2$. Although electrons are allowed to commute between $S_1$ and $S_2$ via nonzero inter-part coupling $\mathbf{h}_{21}$, the occupied band electron's wave function is caged in the $S_1$ region. One of the well-known mechanisms for the stabilization of this type of compactly supported wave function is destructive interference, which cancels all the hopping processes from $S_1$ to $S_2$. As a relevant example, we later examine a topological insulator model on a modified Lieb lattice, demonstrating that the wave functions of the topological bands exhibit compact support as a result of destructive interference.

The constraint (3) implies that every row vector of $\mathbf{h}_{21}(\mathbf{k})$ should be orthogonal to the occupied eigenvectors ($\mathbf{v}_{1,n}^{(o)\dagger}(\mathbf{k})$). Namely, the row vectors of $\mathbf{h}_{21}(\mathbf{k})$ can be constructed by a linear combination of unoccupied eigenvectors $\mathbf{v}_{1,n}^{(u)\dagger}(\mathbf{k})$ because $\mathbf{v}_{1,n}^{(u)\dagger}(\mathbf{k}) \cdot \mathbf{v}_{1,m}^{(o)}(\mathbf{k}) = 0$. However, the linear combination of $\mathbf{v}_{1,n}^{(u)\dagger}(\mathbf{k})$'s is not usually qualified as a Hamiltonian matrix element of a short-ranged hopping model because this is mostly not a finite sum of exponential phase factors, denoted as $e^{i\mathbf{n}\cdot\mathbf{k}}$, where $\mathbf{n}$ is a vector consisting of integer components. Nevertheless, it has been shown that one can obtain an unnormalized eigenvector as a finite sum of Bloch phases in a flat band[51,55]. This can be understood briefly as follows. A general Bloch Hamiltonian $\mathbf{H}(\mathbf{k})$ corresponding to a tight-binding model with a finite hopping range consists of a finite sum of Bloch exponential factors in each element of it. Let us assume that this Hamiltonian hosts a flat band at $E = E_{\text{flat}}$. Then, the eigenvalue equation for the flat band is given by $(\mathbf{H}(\mathbf{k}) - E_{\text{flat}}\mathbf{I})\mathbf{v}_\mathbf{k} = 0$, where the elements of $\mathbf{H}(\mathbf{k}) - E_{\text{flat}}\mathbf{I}$ are still a finite sum of the Bloch phases because $E_{\text{flat}}$ is momentum-independent. Therefore, the components of the vectors in the kernel of $\mathbf{H}(\mathbf{k}) - E_{\text{flat}}\mathbf{I}$ can also be written as the finite sums of the Bloch factors if we allow unnormalized solutions[51]. This special form of the eigenvector explains the existence of compact localized eigenstates for a flat band, which can be obtained

by a Fourier transformation of the unnormalized Bloch eigenvector. Consequently, if there are flat bands among the unoccupied bands, the condition (3) can be satisfied by designing every row of $\mathbf{h}_{12}(\mathbf{k})$ to be a proper linear combination of the eigenvectors of the unoccupied flat bands of $\mathbf{h}_1(\mathbf{k})$. Although we assume the flat bands are unoccupied for clarity, they may also be occupied if they are irrelevant to the system's topology.

## General theory: semimetal case

Secondly, we discuss the protection of a band-crossing attributed to the LSS and destructive interference by establishing a condition for the inter-part interaction $\mathbf{h}_{12}(\mathbf{k})$. Let us consider a crystalline symmetry denoted by $g$. For the $\mathcal{S}_1$ part, the symmetry is represented by an $N_1 \times N_1$ unitary matrix $\mathbf{D}_{1,\mathbf{k}}(g)$. Then, we consider an $N \times N$ unitary matrix representing the LSS operation given by

$$\mathbf{U}(\mathbf{k}) = \begin{pmatrix} \mathbf{D}_{1,\mathbf{k}}(g) & 0 \\ 0 & \mathbf{I}_{N_2} \end{pmatrix}, \tag{4}$$

where an $N_2 \times N_2$ identity matrix $\mathbf{I}_{N_2}$ implies that part $\mathcal{S}_2$ is unaffected by $g$. We denote the eigenvalue of $\mathbf{U}(\mathbf{k})$ by $\lambda_{\mathbf{U},m}(\mathbf{k})$, where $m$ is an integer index representing different eigenvalues. While $\mathbf{H}(g\mathbf{k}) \neq \mathbf{U}_g(\mathbf{k})$ $\mathbf{H}(\mathbf{k})\mathbf{U}_g(\mathbf{k})^\dagger$ due to the broken $g$-symmetry in the whole system, we have $\mathbf{h}_1(g\mathbf{k}) = \mathbf{D}_{1,\mathbf{k}}(g)(\mathbf{k})\mathbf{h}_1(\mathbf{k})\mathbf{D}_{1,\mathbf{k}}(g)^\dagger$, or equivalently,

$$\mathcal{P}_1 \mathbf{H}(g\mathbf{k})\mathcal{P}_1 = \mathcal{P}_1 \mathbf{U}(\mathbf{k})\mathbf{H}(\mathbf{k})\mathbf{U}(\mathbf{k})^\dagger \mathcal{P}_1, \tag{5}$$

where $\mathcal{P}_1$ is a projection operator onto $\mathcal{S}_1$ because the $\mathcal{S}_1$ part respects the symmetry. Conventionally, when a symmetry is respected over the entire system, we expect that a band-crossing between two bands with different symmetry eigenvalues can be protected on a symmetry-invariant line or plane satisfying $g\mathbf{k} = \mathbf{k}$, In this case, no projection operators $\mathcal{P}_1$ are involved between the Hamiltonian and the unitary matrix, unlike Eq. (5) for the case of the LSS. Namely, we usually do not anticipate a symmetry-protected band-crossing from the projected commutation relation (5). Nevertheless, although $\mathbf{H}(g\mathbf{k}) \neq \mathbf{U}_g(\mathbf{k})\mathbf{H}(\mathbf{k})$ $\mathbf{U}_g(\mathbf{k})^\dagger$, $\mathbf{H}(\mathbf{k})$ can commute with $\mathbf{U}(\mathbf{k})$ at least on the symmetry-invariant line or plane (denoted by $\mathbf{k}_g$), namely $\mathbf{H}(\mathbf{k}_g) = \mathbf{U}(\mathbf{k}_g)\mathbf{H}(\mathbf{k}_g)\mathbf{U}(\mathbf{k}_g)^\dagger$ if it satisfies

$$\mathbf{h}_{12}(\mathbf{k}_g) = \mathbf{D}_{1,\mathbf{k}_g}(g)\mathbf{h}_{12}(\mathbf{k}_g). \tag{6}$$

That is, $\mathbf{h}_{12}(\mathbf{k})$ consists of column vectors, which (i) are the eigenvectors of the LSS operator $\mathbf{D}_{1,\mathbf{k}}(g)$ with an eigenvalue 1 on $\mathbf{k} = \mathbf{k}_g$ or (ii) vanish on $\mathbf{k} = \mathbf{k}_g$. Detailed descriptions of both cases are given below.

In both cases (i) and (ii), destructive interference of the Bloch wave function plays a key role in the protection of band-crossings. First, in the case (i), we assume that there are two crossing bands of $\mathbf{H}(\mathbf{k}_g)$ and denote the corresponding eigenvectors as

$$\mathbf{V}_A(\mathbf{k}_g) = \begin{pmatrix} \mathbf{v}_{A,1}(\mathbf{k}_g) \\ \mathbf{v}_{A,2}(\mathbf{k}_g) \end{pmatrix} \text{ and } \mathbf{V}_B(\mathbf{k}_g) = \begin{pmatrix} \mathbf{v}_{B,1}(\mathbf{k}_g) \\ \mathbf{v}_{B,2}(\mathbf{k}_g) \end{pmatrix}, \tag{7}$$

respectively, where $\mathbf{v}_{\alpha,i}(\mathbf{k}_g)(\alpha{=}A,B)$ is an $N_i \times 1$ column vector. $\mathbf{v}_{\alpha,1}(\mathbf{k}_g)$ is a common eigenvector of $\mathbf{D}_{1,\mathbf{k}_g}(g)$ and $\mathbf{h}_1(\mathbf{k}_g)$ due to the LSS in $\mathcal{S}_1$. We denote the eigenvalues of $\mathbf{U}(\mathbf{k}_g)$ and $\mathbf{D}_{1,\mathbf{k}_g}(g)$ corresponding to $\mathbf{V}_\alpha(\mathbf{k}_g)$ and $\mathbf{v}_{\alpha,1}(\mathbf{k}_g)$ by $\lambda_{\mathbf{U},\alpha}$ and $\eta_{1,\alpha}(g)$, respectively. While two bands of $\mathbf{H}(\mathbf{k}_g)$ crossing each other must have different eigenvalues of $\mathbf{U}(\mathbf{k}_g)$, namely $\lambda_{\mathbf{U},A} \neq \lambda_{\mathbf{U},B}$, let us first consider the case where $\lambda_{\mathbf{U},A} = 1$. Since $\mathbf{D}_{1,\mathbf{k}_g}(g)$ is a diagonal block of $\mathbf{U}(\mathbf{k}_g)$, $\lambda_{\mathbf{U},A}$ must be identical to $\eta_{1,A}(g)$ if $\mathbf{v}_{A,1}$ is nonzero. Namely, $\lambda_{\mathbf{U},A} = \eta_{1,A}(g) = 1$. Moreover, any arbitrary vector $\mathbf{v}_{A,2}(\mathbf{k}_g)$ is an eigenvector of an identity matrix $\mathbf{I}_{N_2}$ in Eq. (4) with an eigenvalue 1. Therefore, $\mathbf{v}_{A,2}(\mathbf{k}_g)$ can be nonzero in this case ($\lambda_{\mathbf{U},A} = \eta_{1,A}(g) = 1$). In the case of $\mathbf{v}_{A,1} = 0$, however, it is possible to have $\lambda_{\mathbf{U},A} = 1$, even though $\eta_{1,A}(g) \neq 1$. On the other hand, another eigenvector $\mathbf{V}_B(\mathbf{k}_g)$ with

$\eta_{1,B}(g) \neq 1$, $\mathbf{v}_{B,2}(\mathbf{k}_g)$ must be a zero column vector. If $\mathbf{v}_{B,2}(\mathbf{k}_g)$ is nonzero, $\mathbf{V}_B(\mathbf{k}_g)$ cannot be an eigenvector of $\mathbf{U}(\mathbf{k}_g)$ because the eigenvalues of $\mathbf{D}_{1,\mathbf{k}_g}(g)$ and $\mathbf{I}_{N_2}$ for $\mathbf{v}_{B,1}(\mathbf{k}_g)$ and $\mathbf{v}_{B,2}(\mathbf{k}_g)$ are not identical. If $\mathbf{v}_{B,1}(\mathbf{k}_g)$ is an eigenvector of $\mathbf{h}_1(\mathbf{k}_g)$ with an eigenvalue $\mathcal{E}(\mathbf{k}_g)$, the full eigenvector $\mathbf{V}_B(\mathbf{k}_g)$ with $\mathbf{v}_{B,2}(\mathbf{k}_g) = 0$ is also an eigenvector of $\mathbf{H}(\mathbf{k}_g)$ with the same eigenenergy because every column vector of $\mathbf{h}_{12}(\mathbf{k}_g)$ is an eigenvector of $\mathbf{D}_{1,\mathbf{k}_g}(g)$ with an eigenvalue 1, as described in Eq. (6), which is orthogonal to $\mathbf{v}_{B,1}(\mathbf{k}_g)$. Notably, this implies that the Bloch wave function of at least one of the two crossing bands should have zero amplitudes for orbitals belonging to $\mathcal{S}_2$ for the protection of the band-crossing by a local support symmetry. Otherwise, if $\mathbf{v}_{B,2}(\mathbf{k}_g) \neq 0$ as $\mathbf{v}_{A,2}(\mathbf{k}_g)$ does, both symmetry eigenvalues $\lambda_{\mathbf{U},A}$ and $\lambda_{\mathbf{U},B}$ have to be one simultaneously, and the band-crossing is not allowed. Finally, if $\lambda_{\mathbf{U},A} \neq 1$, $\mathbf{v}_{A,2}(\mathbf{k}_g)$ have to be zero, as in the case of $\mathbf{V}_B(\mathbf{k}_g)$, the bands corresponding to $\mathbf{V}_A$ and $\mathbf{V}_B$ can cross each other if $\lambda_{\mathbf{U},A} \neq \lambda_{\mathbf{U},B}$. As in the previous case, destructive interference can stabilize this type of compactly supported eigenvector, which spans only a part of the system. Note that while we have specified the form of $\mathbf{h}_{12}$ only on $\mathbf{k}_g$, any form of $\mathbf{h}_{12}$ at an arbitrary momentum is allowed as long as it fulfills the condition (6) on $\mathbf{k}_g$.

On the other hand, in case (ii), the coupling between two parts of the system simply disappears at the symmetry-invariant momenta, and the $\mathcal{S}_1$ part is isolated from $\mathcal{S}_2$. Although the crystalline symmetry is not protected in the entire system ($\mathbf{H}(g\mathbf{k}) \neq \mathbf{U}_g(\mathbf{k})\mathbf{H}(\mathbf{k})\mathbf{U}_g(\mathbf{k})^\dagger$), one can find a commutation relation at least at the symmetry-invariant momenta, such that $\mathbf{H}(\mathbf{k}_g) = \mathbf{U}(\mathbf{k}_g)\mathbf{H}(\mathbf{k}_g)\mathbf{U}(\mathbf{k}_g)^\dagger$. As a result, the band-crossing protected in the band structure of $\mathbf{h}_1(\mathbf{k})$ can still be robust even after it couples to $\mathbf{h}_2(\mathbf{k})$, even if the symmetry relevant to the band-crossing of $\mathbf{h}_1(\mathbf{k})$ is broken due to this coupling. Although the coupling in the Bloch Hamiltonian $\mathbf{H}(\mathbf{k})$ between two parts vanishes for $\mathbf{k} = \mathbf{k}_g$, the hopping processes between two parts in real space are nonzero. Under these circumstances, the compactly supported character of these two eigenvectors can be attributed to the destructive interference, as described in the herringbone lattice model example in the subsection dealing with Model-III below.

The band-crossing protected by the LSS can be lifted when the hopping parameters are tuned such that the compactly supported wave function is no longer stabilized. For instance, this occurs when longer-ranged hopping processes are introduced, thereby breaking the condition in Eq. (6). The resulting gap size depends on the detailed structure of the compactly supported wave function, including its spatial shape and amplitude configuration in real space. We quantify the fragility of the band-crossing against such perturbations by the ratio $F = \Delta/t'_{\max}$, where $\Delta$ denotes the induced gap, and $t'_{\max}$ represents the maximum amplitude among the additional hopping terms. Typically, one expects $F \sim 1$, implying that the induced gap is comparable to the energy scale of the perturbative hopping processes. However, the precise value of $F$ depends on the extent to which these additional hoppings violate the destructive interference condition. In an extreme case, $F$ can vanish if the added hoppings still satisfy the destructive interference condition. In general, the band-crossing becomes more robust as $F$ approaches zero.

## Model-I: topological insulator with local support time-reversal symmetry

In this section, we introduce a topological insulator model protected by the local support of time-reversal symmetry instead of the conventional time-reversal symmetry. A spinful case is considered. The lattice structure and the hopping processes are illustrated in Fig. 2a. Lattice sites are labeled alphabetically from A to D with distinct colors. Sites A, B, and C consist of a Lieb lattice, and orbitals at these sites belong to the $\mathcal{S}_1$ part, while orbitals at site D compose the $\mathcal{S}_2$ part. At each site, spin-up and spin-down orbitals exist. The nearest-neighbor hopping processes between the three sites (A, B, and C) in the $\mathcal{S}_1$ part are represented by solid black lines, and the hopping amplitude is 1.

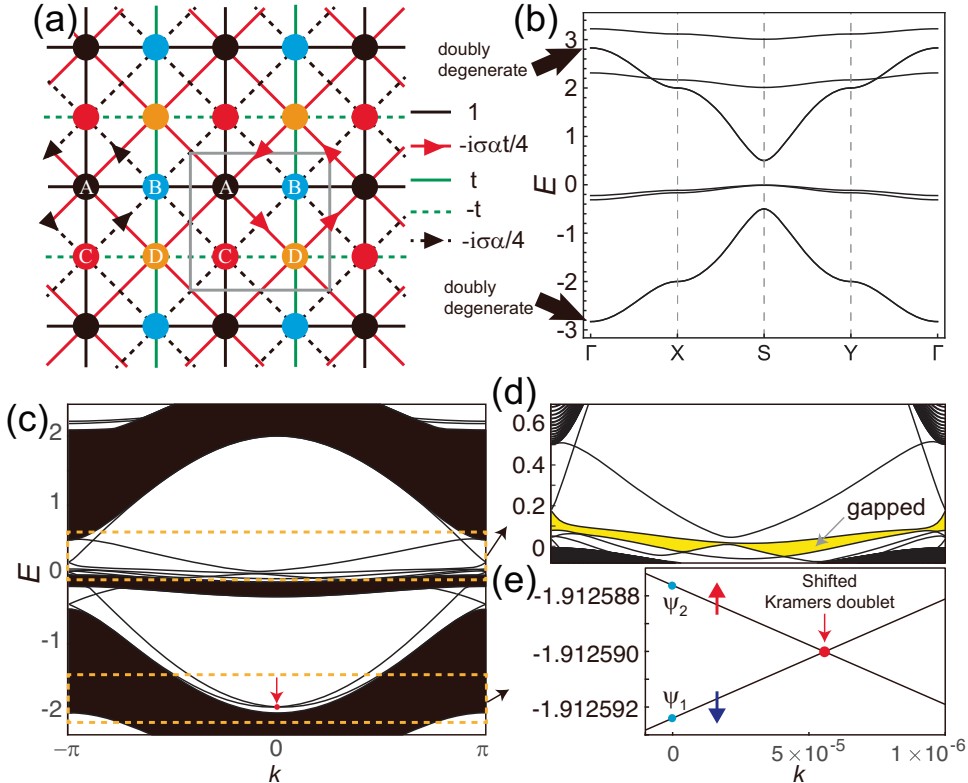

**Fig. 2 | Model-I and local support time-reversal symmetry. a** The lattice and hopping structure of the modified Lieb lattice model. Four sublattices in a unit cell (gray box) are labeled by numbers in the sites. Electrons can hop along the solid and dashed lines, and the corresponding hopping parameters are given on the right-hand side. For the imaginary hopping processes, the direction of the hopping is represented by the arrows. $\alpha$ is the strength of the spin-orbit coupling, and spin-up

and -down are denoted by $\sigma = 1$ and $-1$, respectively. **b** The band structure for $\alpha = 0.5$, $e_\uparrow = 3$, $e_\downarrow = 2$, and $t = 0.3$. Two highly dispersive bands indicated by the arrows are doubly degenerate. **c** The band structure of the ribbon geometry with 100 unit cells along the $x$-direction. The band-crossing enforced by $S_z$ symmetry, slightly away from $k = 0$, is indicated by the red arrow. The edge bands in the two yellow dashed boxes are highlighted in (**d**) and (**e**).

We also consider imaginary-valued hopping processes between sites B and C, induced by the spin-orbit coupling, as indicated by black dashed lines in Fig. 2a. We choose the Lieb lattice model with the spin-orbit coupling for the $S_1$ part because this model possesses the topological and flat bands simultaneously, while we use the flat bands' eigenvectors usefully for the construction of the inter-part coupling.

The $S_1$-part block of the Hamiltonian (1) is given by a $6 \times 6$ matrix $\mathbf{h}_1(\mathbf{k}) = \mathbf{h}_{1,\uparrow}(\mathbf{k}) \oplus \mathbf{h}_{1,\downarrow}(\mathbf{k})$, where

$$\mathbf{h}_{1,\sigma}(\mathbf{k}) = \begin{pmatrix} 0 & 1+e^{-ik_x} & 1+e^{ik_y} \\ 1+e^{ik_x} & 0 & f^*(\mathbf{k}, \sigma\alpha) \\ 1+e^{-ik_y} & f(\mathbf{k}, \sigma\alpha) & 0 \end{pmatrix}, \quad (8)$$

which is obtained in the Bloch basis $\mathbf{C}_{\sigma,\mathbf{k}}^\dagger = (c_{A,\sigma,\mathbf{k}}^\dagger, c_{B,\sigma,\mathbf{k}}^\dagger, c_{C,\sigma,\mathbf{k}}^\dagger)^T$. Here, $f(\mathbf{k}, \alpha) = i\alpha/4 - i\alpha e^{-ik_x}/4 - i\alpha e^{-ik_y}/4 + i\alpha e^{-i(k_x+k_y)}/4$ is the spin-orbit coupling (SOC) and $\sigma = +1(-1)$ for spin $\uparrow(\downarrow)$, so that it has $S_z$ symmetry. This sub-Hamiltonian respects time-reversal symmetry because the sign of the spin-orbit coupling term is opposite for spin-up and down. Using the time-reversal operator given by $\mathbf{T} = i\sigma_y\mathcal{K}$, where $\sigma_y$ is a Pauli matrix for a spin and $\mathcal{K}$ is the complex conjugate operator, one can show that $\mathbf{T}\mathbf{h}_1(-\mathbf{k})\mathbf{T} = \mathbf{h}_1(\mathbf{k})$. Before introducing spin-orbit coupling (SOC), the Hamiltonian $\mathbf{h}_1(\mathbf{k})$ exhibits, for each spin, a flat band at zero energy that touches a Dirac band. Owing to $S_z$ symmetry, all bands are doubly degenerate. Upon turning on the SOC, the Dirac band-crossings become gapped, while the flat bands remain flat, becoming isolated from the upper and lower doubly degenerate dispersive bands. We are interested in the topological properties of the lower dispersive bands below the flat bands. It is well-established that the lower two dispersive bands of this model exhibit a nontrivial $\mathbb{Z}_2$ index,

while the Chern numbers are $+1$ and $-1$ for spin-up and -down, respectively[56,57]. On the other hand, the time-reversal symmetry breaking originates from the $S_2$ part, which consists of D sites. At each D site, one spin-up and one spin-down orbital reside. The sub-Hamiltonians for $S_2$-part is given by

$$\mathbf{h}_2(\mathbf{k}) = \begin{pmatrix} \epsilon_\uparrow & 0 \\ 0 & \epsilon_\downarrow \end{pmatrix}, \quad (9)$$

where $\epsilon_\uparrow$ and $\epsilon_\downarrow$ are the onsite energies for spin-up and down electrons at the D site. As in the case of $\mathbf{h}_1(\mathbf{k})$, $\mathbf{h}_2(\mathbf{k})$ also has $S_z$ symmetry. If $\epsilon_\uparrow \neq \epsilon_\downarrow$, time-reversal symmetry is broken. While we chose the simplest form of $\mathbf{h}_2$, a general form of $\mathbf{h}_2$ does not affect the topological character of the system as long as the added bands due to $\mathbf{h}_2$ lie at high enough energies so that they are unoccupied. Finally, the inter-part block is expressed as a $6 \times 2$ matrix given by

$$\mathbf{h}_{12}(\mathbf{k}) = t\begin{pmatrix} \mathbf{v}_{\mathrm{fb},\uparrow} & O_3 \\ O_3 & \mathbf{v}_{\mathrm{fb},\downarrow} \end{pmatrix}, \quad (10)$$

where $\mathbf{v}_{\mathrm{fb},\sigma} = (-i\sigma\alpha(1 - e^{-ik_x})(1 - e^{ik_y})/4, -(1+e^{ik_y}), 1+e^{-ik_x})^T$ is the unnormalized eigenvector corresponding to the flat band of $\mathbf{h}_1(\mathbf{k})$ and $O_3$ is a $3 \times 1$ zero vector. $t$ is a parameter characterizing the inter-part coupling between $S_1$ and $S_2$. The hopping structure of the entire Hamiltonian $\mathbf{H}(\mathbf{k})$, including the inter-part coupling, is plotted in Fig. 2a. As shown in Fig. 2b, the flat bands are deformed to nearly flat bands, turning on the inter-part coupling.

Most importantly, the inter-part coupling matrix $\mathbf{h}_{12}(\mathbf{k})$ fulfills the condition (3) because its column vectors, being the unnormalized

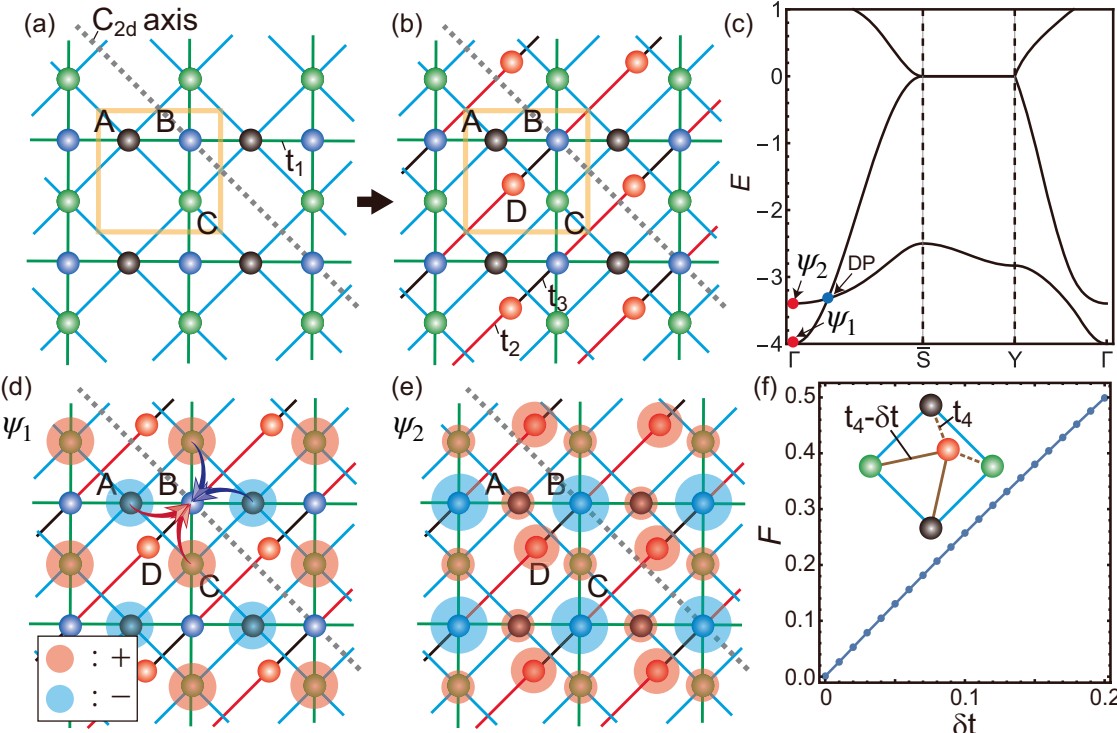

**Fig. 3 | Model-II and local support $C_2$ symmetry. a** The $\mathcal{S}_1$ part of Model-II. The hopping amplitudes between A and C sites are one, while those from B sites to A or C ones are $t_1$. We assume that the onsite energies at A, B, and C sites are identically zero. Therefore, the system preserves $C_2$ symmetry about the axis indicated by a gray dashed line. The yellow box is the unit cell. **b** The entire lattice and hopping structures of Model-II. D sites consist of the $\mathcal{S}_2$ part. The onsite energy at D sites is also zero. The hopping processes from D to B sites are denoted by $t_2$ and $t_3$. **c** The band dispersion for $\{t_1, t_2, t_3\} = \{1.2, 2, 0.5\}$. A Dirac point along $\bar{\Gamma}\bar{S}$, where $\bar{S}$ represents $k = (-\pi, \pi)$, is denoted by DP. Three Bloch states ($\psi_1$ and $\psi_2$) corresponding to red dots are plotted in (**d**) and (**e**). Red and blue circles indicate the positive and negative signs of the wave function, respectively, with their sizes representing its magnitude. $\psi_1$ spans only the sites belonging to $\mathcal{S}_1$ due to the destructive interference indicated by red and blue curved arrows in (**d**). **f** The fragility of the Dirac band-crossing against the additional hopping processes described by $t_4$ and $t_4 - \delta t$.

eigenvectors of the flat bands of $\mathbf{h}_1(\mathbf{k})$, must be orthogonal to the eigenvectors of the lowest two topological bands of $\mathbf{h}_1(\mathbf{k})$, denoted by $\mathbf{v}_{1,1}^{(o)}$ and $\mathbf{v}_{1,2}^{(o)}$. As mentioned above, the Chern numbers calculated using these two eigenvectors are 1 and −1, respectively. Turning on $\mathbf{h}_{12}$, satisfying the condition (3), the band spectra of these topological bands corresponding to $\mathbf{v}_{1,1}^{(o)}$ and $\mathbf{v}_{1,2}^{(o)}$ are unaffected, and the eigenvector can be expressed as Eq. (2). The amplitudes of $\mathbf{v}_{1,1}^{(o)}$ and $\mathbf{v}_{1,2}^{(o)}$ at A, B, and C sites cancel each other at D sites after the hopping processes via destructive interference. Consequently, the topological characteristics of those two bands remain unchanged both before and after the incorporation of $\mathbf{h}_{12}$. Note that those topological bands, indicated by the lower arrow in Fig. 2b, are degenerate due to the coexistence of the local support time-reversal and inversion symmetries.

We investigate the edge states by considering a ribbon geometry translationally invariant along $y$-direction, as shown in Supplementary Fig. 3a. Here, we apply an onsite potential to the sites near the edges of the ribbon to make the crossing point between edge-localized bands completely detached from the bulk band continuum and more visible. In the gap just above the lowest bulk bands, which exhibits a nontrivial $\mathbb{Z}_2$ index, boundary modes emerge that cross each other and connect the bulk conduction and valence bands. The red dot indicates the band-crossing of the edge-localized bands. On the other hand, the edge-localized dispersions within the gap right above the nearly flat bands, indicated by the upper yellow dashed box in Fig. 2c, do not exhibit such connectivity between the bulk bands, as highlighted by the yellow gap region in Fig. 2d, separating upper and lower bands completely. Note that the Bloch wave functions in the nearly flat bands span both $\mathcal{S}_1$ and $\mathcal{S}_2$ parts, breaking time-reversal symmetry, while those in the lowest two bulk bands occupy only the time-reversal

symmetric $\mathcal{S}_1$ part. The breaking of time-reversal symmetry is reflected in the lower edge bands in the lower yellow box in Fig. 2c. The band-crossing point is slightly shifted from the time-reversal invariant momentum, as highlighted in Fig. 2e. Although the lowest two bulk bands lack contributions from the D-site orbitals, the edge modes contain a very small amount, as shown in Supplementary Fig. 3f. As a result, Kramer's degeneracy is not guaranteed for the edge-localized bands even though the gap is characterized by the nontrivial $\mathbb{Z}_2$ invariant. Still, the band-crossing remains robust because two crossing bands belong to different spin species, namely, it is enforced by $S_z$ symmetry. This kind of decoupling of spin-up and spin-down sectors is possible because the SOC considered in our model does not flip the spin, as in the case of the Kane-Mele quantum spin Hall insulator model on graphene. The shift of the crossing point is extremely small, as the contribution of D-site orbitals is negligibly low, as shown in Supplementary Fig. 3f. By replacing Δ in the definition of the fragility with the energy splitting of the edge states at $k = 0$, the fragility of the helical edge modes is estimated to be $F \approx 1.57 \times 10^{-5}$. If we allow spin-flipping processes in $\mathbf{h}_2$, a small gap opens between the two edge bands. Namely, although the occupied bands possess a nontrivial $\mathbb{Z}_2$ topology, Kramer's degeneracy can be broken in the edge-localized bands.

## Model-II: Dirac fermions protected by local support $C_2$ symmetry

This section considers a four-site square lattice model exhibiting Dirac nodes protected by the local support $C_2$ symmetry. Its lattice and hopping structures are illustrated in Fig. 3a and b. The entire system is described in Fig. 3b, where the unit cell comprises four sites, each labeled A to D, with one orbital assigned to each site. The system is

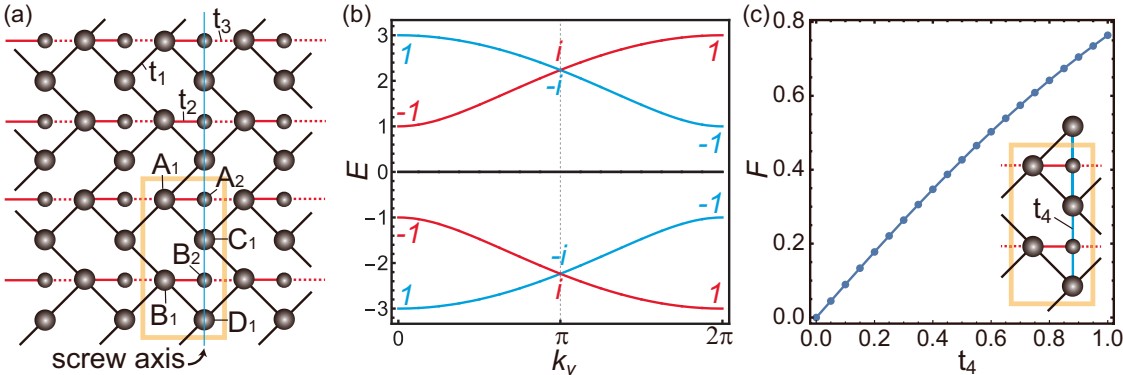

**Fig. 4 | Model-III and local support nonsymmorphic symmetry. a** The lattice and hopping structures of the herringbone lattice model. In the unit cell, indicated by a yellow box, there are six sublattice sites denoted by $A_1$, $B_1$, $C_1$, and $D_1$ belonging to the $\mathcal{S}_1$ part and $A_2$ and $B_2$ belonging to the $\mathcal{S}_2$ part. Hopping processes with amplitudes $t_1$, $t_2$, and $t_3$ are expressed by black and red lines. The blue line represents the screw axis considered in our model. **b** The band structure along $k_y$-axis

($k_x = 0$) for the band parameters $\{t_1, t_2, t_3\} = \{1, 1, -1\}$. There are doubly degenerate flat bands at zero energy. The eigenvalues of the screw operation $\{C_{2y}|\frac{1}{2}\hat{y}\}$ for each band are represented by the complex numbers with the same color as the corresponding band. **c** The fragility of the band-crossing at $k_y = \pi$ in (**b**), as a function of the additional hopping processes represented by $t_4$.

---

divided into $\mathcal{S}_1$ (A, B, and C sites) and $\mathcal{S}_2$ (D sites). The $\mathcal{S}_1$ part is drawn in Fig. 3a. The $C_2$ symmetry of our concern is about the $d$-axis indicated by the gray dashed line in Fig. 3a, and we call it a $C_{2d}$ symmetry. We assume that the $\mathcal{S}_1$ part respects $C_{2d}$ symmetry, which is the case when the onsite energies at A and C sites, denoted by $\epsilon_A$ and $\epsilon_C$, are identical. However, the entire system does not possess $C_{2d}$ symmetry because the $\mathcal{S}_1$ part couples to the $\mathcal{S}_2$ part, which breaks $C_{2d}$ symmetry as the intra-cell and inter-cell hopping amplitudes between B and D sites are distinct. The Hamiltonian (1) corresponding to this model is composed as

$$\mathbf{h}_1(\mathbf{k}) = \begin{pmatrix} 0 & 1+e^{-ik_x}+e^{ik_y}+e^{-i(k_x-k_y)} & t_1(1+e^{-ik_x}) \\ 1+e^{ik_x}+e^{-ik_y}+e^{i(k_x-k_y)} & 0 & t_1(1+e^{-ik_y}) \\ t_1(1+e^{ik_x}) & t_1(1+e^{ik_y}) & 0 \end{pmatrix},$$

(11)

$\mathbf{h}_2(\mathbf{k}) = 0$, and $\mathbf{h}_{21}(\mathbf{k}) = (0, 0, t_2 + t_3 e^{i(k_x+k_y)})$. We design the explicit form of the inter-part coupling by letting its columns satisfy the condition (6) of the general theory. We first find the unitary matrix for the local support $C_{2d}$ symmetry and then obtain its eigenvector with eigenvalue 1. Then, any column vectors proportional to such an eigenvector compose a proper inter-part coupling matrix hosting a band-crossing protected by local support $C_{2d}$ symmetry, assisted by destructive interference.

As shown in Fig. 3c, a band-crossing, denoted by DP, exists along $\Gamma\bar{S}$ when $\epsilon_A = \epsilon_C$, where $\bar{S}$ stands for $(-\pi, \pi)$. While we usually expect that $C_{2d}$ symmetry for an axis parallel to $\Gamma\bar{S}$ would protect this band-crossing, such symmetry is absent in the entire system because the hopping amplitudes between the intra-cell (black lines) and inter-cell (red lines) bondings between B and D sites are different from each other. We can understand the protective mechanism for this band-crossing as follows. The unitary matrix corresponding to the local support $C_{2d}$ symmetry of the $\mathcal{S}_1$ part is given by

$$\mathbf{D}_{1,C_{2d}}(\mathbf{k}) = \begin{pmatrix} 0 & e^{ik_y} & 0 \\ e^{ik_x} & 0 & 0 \\ 0 & 0 & 1 \end{pmatrix},$$

(12)

which satisfies $\mathbf{D}_{1,C_{2d}}(\mathbf{k})\mathbf{h}_1(\mathbf{k})\mathbf{D}_{1,C_{2d}}^\dagger(\mathbf{k}) = \mathbf{h}_1(C_{2d}\mathbf{k})$. Let us denote the momentum parallel(perpendicular) to $\Gamma\bar{S}$ by $k_1(k_2)$. Namely, $k_1 = (k_x + k_y)/\sqrt{2}$ and $k_2 = (-k_x + k_y)/\sqrt{2}$. On the $C_{2d}$-invariant momentum $k_2$ ($k_1 = 0$), one of the eigenvectors of $\mathbf{D}_{1,C_{2d}}$ with an eigenvalue 1 is given by $\mathbf{v}_1(k_2) = (0, 0, 1)^T$. Therefore, any vectors

proportional to $\mathbf{v}_1(k_2)$ can build the inter-part coupling $\mathbf{h}_{12}$ satisfying the condition (6) along the $C_{2d}$-invariant momentum. In the model, we choose $\mathbf{h}_{21}(\mathbf{k}) = (0, 0, t_2 + t_3 e^{i(k_x+k_y)})$. One can also check that the full Hamiltonian commutes with the full unitary matrix $\mathbf{U}(\tilde{k}_2)$ given in the form (4) when $k_1 = 0$. Indeed, the bands crossing at DP in Fig. 3c have opposite signs of the eigenvalues of $\mathbf{D}_{1,C_{2d}}(k_2)$ or $\mathbf{U}(\tilde{k}_2)$. Some example Bloch wave functions are illustrated in Fig. 3d and e. Due to the lack of $C_{2d}$ symmetry in the entire system, $\Psi_2$ is not $C_{2d}$ symmetric, as shown in Fig. 3(e). However, $\Psi_1$ can still be an antisymmetric eigenstate of $C_{2d}$ operation despite the breaking of $C_{2d}$ symmetry because it spans only the $\mathcal{S}_1$ part, which respects $C_{2d}$ symmetry. $\Psi_1$ can exhibit such a local support property because its amplitudes at A and C sites cancel each other precisely when they hop to the neighboring B sites due to the destructive interference as described by the curved arrows in Fig. 3d. Therefore, the protection of the band-crossing phenomenon is attributed to the local support $C_{2d}$ symmetry within $\mathcal{S}_1$. The condition for destructive interference can be violated by introducing additional hopping processes from the D site to its neighboring A and C sites, as illustrated in the inset of Fig. 3f. When the hopping amplitudes for these four processes are identical ($\delta t = 0$), the destructive interference remains intact because the amplitudes arriving at the D site from the A and C sites cancel each other. In contrast, when $\delta t \neq 0$, this cancellation no longer occurs, and the band-crossing becomes gapped. In Fig. 3f, we quantify the fragility of the band-crossing as a function of $\delta t$. One finds that the induced band gap remains strongly suppressed for small values of $\delta t$, indicating the robustness of the crossing against the longer-range hopping processes.

## Model-III: Dirac fermions enforced by local support nonsymmorphic symmetry

In this section, we consider an example model with band-crossings enforced by a local support nonsymmorphic symmetry. The lattice structure is illustrated in Fig. 4a, which is called a modified herringbone lattice model. Each unit cell contains six sites, which are divided into two subsystems: $\mathcal{S}_1$, comprising $A_1$, $B_1$, $C_1$, $D_1$, and $\mathcal{S}_2$, comprising $A_2$, $B_2$. There are three types of hopping processes, indicated by black solid, red solid, and red dashed lines, with amplitudes $t_1$, $t_2$, and $t_3$, respectively. The model for the $\mathcal{S}_1$ part is the herringbone lattice model[58]. While the herringbone model in $\mathcal{S}_1$ part respects two screw symmetries, one of the screw axes of our interest is denoted by the blue line. About this axis, $\mathcal{S}_1$ part preserves an off-centered nonsymmorphic rotation symmetry $\{C_{2y}|\frac{1}{2}\mathbf{a}_y\}$, which represents a two-fold rotation followed by a partial translation $\mathbf{a}_y/2$. By coupling $\mathcal{S}_1$ part with

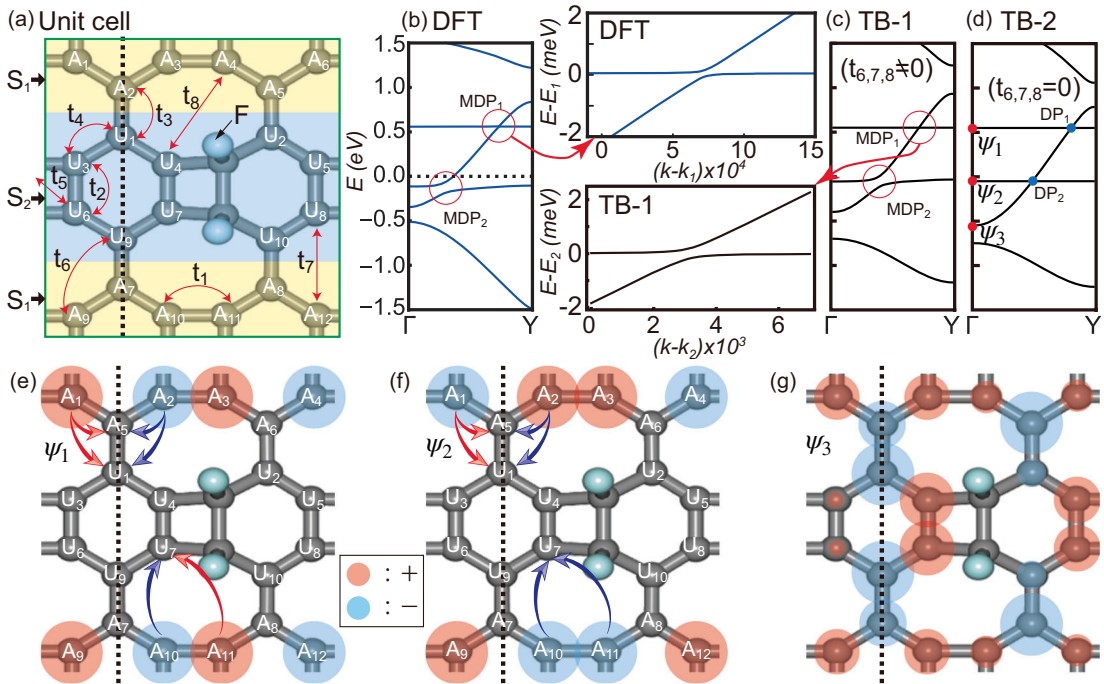

**Fig. 5 | Fluorinated biphenylene network. a** A unit cell (green box) of the biphenylene network passivated by two fluorine atoms. Gray (sky blue) spheres represent carbon(fluorine) atoms. Hopping processes are illustrated by red arrows. The unit cell is divided into two parts, $S_1$ and $S_2$. The carbon sites in $S_1$ and $S_2$ are indicated by $A_i$ and $U_i$, respectively. **b** The DFT band structure. Along $\Gamma Y$, two massive type-II Dirac fermions appear at distinct energy levels, labeled as $MDP_i$. The $MDP_1$ feature in the DFT band structure is highlighted in the right-hand panel. $k_1$ is an arbitrary momentum near the $MDP_1$. **c** The tight-binding(TB) band dispersion calculated using the hopping parameters $\{t_1, t_2, t_3, t_4, t_5, t_6, t_7, t_8\} = \{-3.4, -3.1, -3.1,$ $-3.2, -0.8, -0.06, 0.1725, 0.17\}$. The $MDP_1$ of the tight-binding result is enlarged in the left-hand panel, where $k_2$ is an arbitrary momentum near the $MDP_2$. **d** The tight-binding band spectrum computed by suppressing the perturbative hopping terms associated with $t_6$, $t_7$, and $t_8$. Several Dirac nodes ($DP_i$'s) are marked with blue dots. **b**–**d** share the same $y$-axis. From (**e**–**g**), we plot Bloch wave functions $\Psi_i$'s indicated by red dots in (**d**). Red and blue circles denote the positive and negative signs of the wave function, respectively, while the circle size reflects the magnitude of the wave function. The gray dashed vertical lines are the local support mirror symmetry axes. Destructive interferences are described by curved arrows.

$S_2$ one via $t_2$ and $t_3$, the entire system loses the screw symmetry $\{C_{2y}|\frac{1}{2}\mathbf{a}_y\}$.

The Bloch Hamiltonian for $S_1$ part is given by

$$\mathbf{h}_1(\mathbf{k}) = t_1 \begin{pmatrix} 0 & 0 & 1+e^{-ik_x} & e^{ik_y} \\ 0 & 0 & e^{-ik_x} & 1+e^{-ik_x} \\ 1+e^{ik_x} & e^{ik_x} & 0 & 0 \\ e^{-ik_y} & 1+e^{ik_x} & 0 & 0 \end{pmatrix}, \quad (13)$$

and that of $S_2$ part is a $2 \times 2$ zero matrix. The coupling between these two parts is given by

$$\mathbf{h}_{12}(\mathbf{k}) = \begin{pmatrix} t_2 + t_3 e^{ik_x} & 0 & 0 & 0 \\ 0 & t_2 + t_3 e^{ik_x} & 0 & 0 \end{pmatrix}^{\mathsf{T}}. \quad (14)$$

The symmetry-invariant momenta of the screw symmetry $\{C_{2y}|\frac{1}{2}\mathbf{a}_y\}$ is described by $\mathbf{k}_g = (0, k_y)$. Along $\mathbf{k}_g$, the four bands of $\mathbf{h}_1(\mathbf{k})$ exhibit two band-crossings, one between the lowest two bands and another between the highest two bands, at $k_y = \pi$. We note that these band-crossings are enforced by the screw symmetry of $S_1$ part because the screw symmetry eigenvalues of these bands are evaluated as $\pm e^{ik_y/2}$, as shown in Supplementary Information.

Now, we turn on the inter-part couplings $t_2$ and $t_3$, which break the screw symmetry $\{C_{2y}|\frac{1}{2}\mathbf{a}_y\}$. The band dispersion for $t_1 = t_2 = -t_3 = 1$ is plotted in Fig. 4b. One can note that there are still two band-crossings between red and blue bands at $k_y = \pi$. These are protected by the local support screw symmetry, corresponding to the case-(ii) of the semimetallic part of the general theory in the previous section. Let us

denote the $n$-th eigenvector of $\mathbf{h}_1(\mathbf{k}_g)$ as $\mathbf{v}_{n,1}(\mathbf{k}_g)$, which is a $4 \times 1$ column vector. Here, $n$ runs from 1 to 4. Since $\mathbf{h}_{12}(\mathbf{k})$, the coupling between two parts, vanishes on $\mathbf{k}_g$, one can find four eigenvectors of the full Hamiltonian $\mathbf{H}(\mathbf{k}_g)$ in the form $\mathbf{V}_n(\mathbf{k}_g) = (\mathbf{v}_{n,1}(\mathbf{k}_g)^{\mathsf{T}}, \mathbf{v}_{n,2}(\mathbf{k}_g)^{\mathsf{T}})^{\mathsf{T}}$, where $\mathbf{v}_{n,2}(\mathbf{k}_g)$ is a $2 \times 1$ zero vector. These four eigenvectors correspond to the red and blue bands in Fig. 4b. The other two eigenvectors of $\mathbf{H}(\mathbf{k}_g)$ are also denoted by $\mathbf{V}_n(\mathbf{k}_g)$, but with $n = 5$ and 6, corresponding to the doubly degenerate zero energy flat bands in Fig. 4b. Since the four eigenstates ($1 \le n \le 4$) occupy only sites belonging to $S_1$, they exhibit compactly supported character. These states can be stabilized due to destructive interference because their amplitudes at $A_1$ and $B_1$ sites are uniform along the $x$-direction and the hopping parameters toward $A_2$ and $B_2$ sites from the left and right neighbors have opposite signs. Because the Bloch wave functions corresponding to $\mathbf{v}_{n,1}(\mathbf{k}_g)$ and $\mathbf{V}_n(\mathbf{k}_g)$ are exactly the same, they share the same eigenvalues of $\{C_{2y}|\frac{1}{2}\mathbf{a}_y\}$ as indicated by complex numbers in Fig. 4b. One can ruin the destructive interference and open a gap by including the hopping processes from $A_2$ and $B_2$ sites to the neighboring $C_1$ and $D_1$ sites, as illustrated in the inset of Fig. 4c. The corresponding fragility is plotted as a function of $t_4$ in Fig. 4c. One can note that the gap is quite robust for small $t_4$.

## Realistic example: fluorinated biphenylene network

As a realistic example, we consider a fluorinated biphenylene network[54]. The pristine biphenylene network has received great attention because it was the first experimental realization of the non-benzenoid carbon allotrope[59–63]. Unlike graphene, the biphenylene network hosts a type-II Dirac fermion, whose Dirac node is protected by $C_{2y}$ symmetry[64,65]. By attaching two fluorine atoms per unit cell, as shown in Fig. 5a, we break the $C_{2y}$ symmetry. We focus on the

electronic structure along $\Gamma Y$, which comprises symmetry-invariant momenta with respect to $C_{2y}$, as shown in Fig. 5b. Given that $C_{2y}$ symmetry is broken, band-crossings are not expected to be protected by this symmetry. Indeed, we observe massive Dirac dispersions, labeled MDP$_1$ and MDP$_2$. Interestingly, the associated gap sizes are remarkably small compared to the energy scales of the dominant hopping processes ($t_1$ and $t_4$) directed toward the symmetry-breaking fluorine sites from adjacent carbon atoms. In the tight-binding analysis replicating the DFT results, we use $t_1 = -3.4$ eV and $t_4 = -3.2$ eV. Notably, the gap of MDP$_1$ is exceptionally small, on the order of sub-millielectronvolts, as highlighted in the right-hand panel of Fig. 5b. Using tight-binding analysis, we demonstrate below that these two MDPs originate from local support $C_{2y}$ symmetry about the $y$-axis (the dashed line in Fig. 5a) and destructive interference.

In Fig. 5a, the unit cell is represented by the green box, and it is divided into two parts, denoted by $S_1$ (yellow region) and $S_2$ (blue region). The carbon sites belonging to $S_1$ and $S_2$ are labeled by A$_i$ and U$_i$, respectively, as illustrated in Fig. 5a. Due to the fluorine atoms in $S_2$, $C_{2y}$ symmetry is respected only in $S_1$ while broken over the entire system. The lattice and hopping structures of the fluorinated biphenylene network are illustrated in Fig. 5a. We can regard the fluorinated carbon atoms as vacancy sites so that there are 22 $p_z$-orbitals per unit cell. The corresponding $22 \times 22$ Bloch Hamiltonian matrix $\mathbf{H}(\mathbf{k})$ of the form (1) with a $12 \times 12$ sub-Hamiltonian $\mathbf{h}_1(\mathbf{k})$ for $S_1$, a $10 \times 10$ sub-Hamiltonian $\mathbf{h}_2(\mathbf{k})$ for $S_2$, and the $12 \times 10$ inter-part coupling $\mathbf{h}_{12}(\mathbf{k})$ is given in Section II of Supplementary Information. With the tight-binding parameters $\{t_1, t_2, t_3, t_4, t_5, t_6, t_7, t_8\} = \{-3.4, -3.1, -3.1, -3.2, -0.8, -0.06, 0.1725, 0.17\}$, we obtain the band spectrum in Fig. 5c, which reproduces the DFT result in Fig. 5b. This model is denoted by TB-1. As shown in the lower panel on the left side of Fig. 5c, the gap of MDP$_1$ is extremely small (sub-millielectronvolt), whereas that of MDP$_2$ is around 0.1 eV, consistent with the DFT results.

To understand the origin of the small gaps in MDP$_1$ and MDP$_2$ of the DFT and TB-1 model, we introduce another tight-binding model (TB-2), in which the relatively small hopping parameters $t_6$, $t_7$, and $t_8$ are turned off. These hopping terms in the model TB-1 are then treated as perturbative corrections to the TB-2 model. The band structure of the unperturbed TB-2 model is plotted in Fig. 5d. There are two Dirac band-crossings denoted by DP$_1$ and DP$_2$, although $C_{2y}$ symmetry is still broken due to the attached fluorine atoms. The MDP$_i$ of the TB-1 model and DFT is obtained by opening a gap in the DP$_i$ of the TB-2 model. It is therefore essential to analyze what protects the band-crossing DP$_i$ in the TB-2 model, why the perturbative long-range hopping processes lead to gap openings, and why the gap at DP$_1$ is particularly small.

One can understand how the band-crossings DP$_1$ and DP$_2$ of the TB-2 model on $\Gamma Y$ are not lifted despite the broken $C_{2y}$ symmetry from the real-space picture. In Fig. 5e–g, we draw Bloch wave functions, denoted by $\Psi_1$, $\Psi_2$, and $\Psi_3$, corresponding to the three crossing bands along $\Gamma Y$ in Fig. 5d. $\Psi_1$ and $\Psi_2$ are eigenstates of the $C_{2y}$ operator with an eigenvalue $-1$. Although the full system does not preserve $C_{2y}$ symmetry, a $C_{2y}$-symmetric wave function can still be stabilized because it is supported entirely within the $S_1$ region, which itself respects $C_{2y}$ symmetry. Namely, $\Psi_1$ and $\Psi_2$ exhibit compact support within $S_1$, with vanishing amplitudes in $S_2$. This behavior arises because the amplitudes confined to $S_1$ do not leak into $S_2$ under the allowed hopping processes, owing to destructive interference. For example, the amplitudes of $\Psi_1$ and $\Psi_2$ at A$_1$ and A$_2$ sites cancel each other after hopping to A$_5$ or U$_1$ sites, as indicated by red and blue arrows in Fig. 5e and f. On the other hand, another wave function $\Psi_3$, spanning all sites in the entire system, is not $C_{2y}$-symmetric, as illustrated in Fig. 5e. Therefore, one cannot assign a $C_{2y}$ eigenvalue for $\Psi_3$, unlike the case of $\Psi_1$ and $\Psi_2$. However, $\Psi_3$ is also $C_{2y}$-symmetric projecting onto $S_1$ part, as shown in Fig. 5g. Focusing on the $S_1$ region, the $C_{2y}$ eigenvalue of $\Psi_3$ can be regarded as 1.

We find a unitary matrix $\mathbf{U}(\mathbf{k})$ of the form (4) for the local support $C_{2y}$ symmetry, consisting of a unitary matrix $\mathbf{u}_1(\mathbf{k})$ corresponding to the $C_{2y}$ operation in $S_1$ and an identity matrix for $S_2$ [see Section II of Supplementary Information]. One can show that $\mathbf{u}_1(\mathbf{k})\mathbf{h}_1(k_x, k_y)$ $\mathbf{u}_1(\mathbf{k})^\dagger = \mathbf{h}_1(-k_x, k_y)$ but $\mathbf{U}(\mathbf{k})\mathbf{H}(k_x, k_y)\mathbf{U}(\mathbf{k})^\dagger \neq \mathbf{H}(-k_x, k_y)$, where $\mathbf{h}_1$ is the local Hamiltonian for the $S_1$-part and $\mathbf{H}$ is the Hamiltonian for the entire system. The $\Gamma Y$ line comprises crystal momentum points that are invariant under the local support mirror symmetry, and the momentum along this line is denoted by $\mathbf{k}_{C_2} = (0, k_y)$. On $\Gamma Y$ we show that $\mathbf{U}(\mathbf{k})$ $\mathbf{H}(0, k_y)\mathbf{U}(\mathbf{k})^\dagger = \mathbf{H}(0, k_y)$. The local support $\mathbf{U}(\mathbf{k})$ were able to satisfy this symmetry relation at least along $\Gamma Y$, although $C_{2y}$ symmetry is broken because every column vector of $\mathbf{h}_{12}(\mathbf{k}_{C_2})$ becomes an eigenvector of $\mathbf{u}_1(\mathbf{k}_{C_2})$ with an eigenvalue $\lambda_{C_2} = 1$, satisfying the condition in Eq. (6). Because of the identity matrix in the $S_2$ block of $\mathbf{U}(\mathbf{k})$, the eigenvalue of $\mathbf{U}(\mathbf{k}_{C_2})$ should be one if the corresponding Bloch wave function spans both $S_1$ and $S_2$ parts like $\Psi_3$. However, other Bloch wave functions, whose eigenvalue of $\mathbf{U}(\mathbf{k}_{C_2})$ is not one, must have vanishing amplitudes in $S_2$ region since the eigenvalue of an identity matrix is always one, as in the case of $\Psi_1$ and $\Psi_2$. Namely, these two wave functions are extended along the $x$-axis while compactly localized along the $y$-axis, as shown in Fig. 5e and f. For the stabilization of such eigenfunctions, destructive interference plays a crucial role as explained above. In fact, the existence of such directionally compactly supported eigenfunctions is due to the fact that they are eigenstates of one-dimensional flat bands along $\Gamma Y$. In general, a flat band hosts a compact localized eigenstate having zero amplitudes outside a specific region in real space. One can consider $\mathbf{H}(0, k_y)$ as an effective one-dimensional Hamiltonian of a system translationally invariant along the $y$-axis, hosting two flat bands In real space, this effective one-dimensional model can be interpreted as a nanotube formed by cutting the system into a ribbon of the same width as the unit cell shown Fig. 5a, and then connecting the sites A$_1$, U$_3$, U$_6$, and A$_9$ to A$_6$, U$_5$, U$_8$, and A$_{12}$, respectively. According to the general theory of flat bands[51], these flat bands should have compactly localized eigenmodes along the $y$-axis. $\Psi_1$ and $\Psi_2$ are the linear combinations of these compact localized states, and this is why they possess such a local support characteristic. If we include longer-ranged hopping processes, the column vectors of $\mathbf{h}_{12}(\mathbf{k}_{C_2})$ are no longer the eigenvectors of $\mathbf{u}_1(\mathbf{k}_{C_2})$, and the band-crossing is not protected anymore, as shown in Fig. 5c. The fragilities of the two band-crossings, DP$_1$ and DP$_2$, of the unperturbed tight-binding model shown in Fig. 5d against longer-ranged hopping processes $t_6$, $t_7$, and $t_8$ are evaluated to be $F = 0.00219$ and $0.678$, respectively. The exceptional robustness of DP$_1$ originates from the fact that the destructive interference of $\Psi_1$ is approximately preserved even after the inclusion of $t_6$, $t_7$, and $t_8$, since its amplitudes at A$_{10}$ and A$_{11}$ have opposite signs and the magnitudes of $t_7$ and $t_8$ are comparable. In contrast, for $\Psi_2$, the amplitudes at A$_{10}$ and A$_{11}$ have the same sign, and as a result, the destructive interference is strongly disrupted.

## Discussion

Destructive interference plays a crucial role in LSS protection, where one portion of the divided system remains fixed while a local support symmetry operation is applied to the other. In the scenario involving an isolated topological band, the topological characteristics of $\mathbf{h}_1(\mathbf{k})$ persist even upon its coupling with symmetry-breaking $\mathbf{h}_2(\mathbf{k})$ facilitated by the compactly supported nature of the wave function, made possible by the presence of destructive interference. In protecting the band-crossing, the amplitudes of a wave function must be rigorously zero to uphold the non-unity eigenvalue of the LSS, which is applied to only a part of the system, while the remaining part is unchanged. While we have examined this mechanism primarily in two-dimensional examples, it is equally applicable to three-dimensional systems, including Dirac and nodal-line semimetals. Dirac points and nodal lines, typically protected by crystalline symmetries such as inversion,

mirror, or rotation, can still appear within our framework when these symmetries act locally on the $S_1$ subsystem and the inter-part coupling satisfies the condition in Eq. (6). As a concrete illustration, Supplementary Section V demonstrates that a nodal line in a three-dimensional model can be protected by a local support mirror symmetry. In contrast, Weyl nodes fall outside the primary scope of LSS protection, as their stability originates from a topological monopole charge rather than a crystalline symmetry. Nevertheless, if the inter-part hopping respects the LSS compatibility condition, destructive interference may still confine the Weyl-cone wave functions to $S_1$, thereby constraining how opposite-chirality Weyl nodes hybridize or annihilate through couplings between the $S_1$ and $S_2$ parts. Understanding the role of coupling between block Hamiltonians in shaping topological phases has been a recurring theme in the study of symmetry-protected and crystalline topological matter. A prominent example is the coupled-layer construction, in which couplings between stacked layers give rise to higher-dimensional topological phases[66]. In these approaches, coupling plays a constructive role by generating a global topological obstruction in the full Bloch Hamiltonian. By contrast, the LSS framework focuses on identifying conditions under which coupling fails to destroy pre-existing symmetry-protected features.

Our theory provides a unifying framework for interpreting protective mechanisms underlying a broad class of topological properties and band-crossings in band structures, including situations where the relevant symmetry is preserved only on a restricted subspace. To identify real systems that fall into this category, we may focus on those exhibiting local support symmetry (LSS) and examine whether they satisfy the conditions given in Eq. (3) or (6). Drawing on insights from Model-I and the fluorinated biphenylene network, good candidates among systems with LSSs are those that host flat bands, either across the entire Brillouin zone or within specific regions, because the compactly supported nature of Bloch wave functions in flat bands may facilitate the fulfillment of the aforementioned conditions. As the conditions for destructive interference are challenging to rigorously fulfill in real materials, the protection of topological properties or band-crossings by LSSs may not be strictly ensured. Nevertheless, the breakdown of destructive interference typically occurs through long-range hopping processes, which are generally much smaller than the nearest ones. As a result, the gap-opening effect at the band-crossing is relatively small. Hence, one can still reasonably apply the protection mechanism of the LSS to such systems in an approximate manner, as shown in the fluorinated biphenylene network example.

We note a conceptual relation to the recently introduced notion of sub-symmetry protection, where a symmetry preserved only within a restricted subspace can protect certain boundary zero modes even after the bulk topological invariant is destroyed[67]. In that case, protection arises because the boundary state is fully supported on the subspace on which the sub-symmetry acts, so that symmetry-preserving perturbations do not shift its eigenvalue. By contrast, the LSS framework developed here protects topological features themselves through algebraic constraints on the Hamiltonian, and the destructive interference plays an essential role.

## Methods

### Details of first-principles calculations

The density functional theory (DFT) calculations are performed using the Vienna Ab initio Simulation Package (VASP)[68,69]. The exchange-correlation energy functional was treated with the PBE functional[70], and van der Waals (vdW) interactions between particles were accounted for using the DFT-D3 method[71,72]. For self-consistent calculations, we used a kinetic energy cutoff of 500 eV. We employed k-point sampling grids of $6 \times 6 \times 1$ and $24 \times 24 \times 1$ for structure relaxations and band calculations, respectively[73,74]. The geometry

optimization is carried out until the energy convergence threshold of $10^{-6}$ eV is reached, and the Hellmann-Feynman forces acting on each atom are maintained below 0.001 eV/Å[75].

## Data availability

No new datasets were generated or analyzed during the current study. All data supporting the findings of this study are available from the corresponding author upon request.

## Code availability

The code used in this study is available from the corresponding author upon request.

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

## Acknowledgements

J.W.R. was supported by the National Research Foundation of Korea (NRF) Grant funded by the Korean government (MSIT) (Grant no. 2021R1A2C1010572 and RS-2023-NR068116) and the Ministry of Education (Grant no. RS-2023-00285390). B.A.B. was supported by the Gordon and Betty Moore Foundation through Grant No. GBMF8685 towards the Princeton theory program, the Gordon and Betty Moore Foundation's EPiQS Initiative (Grant No. GBMF11070), the Office of Naval Research (ONR Grant No. N00014-20-1-2303), the Global Collaborative Network Grant at Princeton University, the Simons Investigator Grant No. 404513, the BSF Israel US foundation No. 2018226, the NSF-MERSEC (Grant No. MERSEC DMR 2011750), Simons Collaboration on New Frontiers in Superconductivity (SFI-MPS- NFS-00006741-01), and the Schmidt Foundation at the Princeton University, European Research Council (ERC) under the European Union's Horizon 2020 research and innovation program (Grant Agreement No. 101020833). S.K. was supported by the National Research Foundation (NRF) of Korea (Grant no. RS-2024-00410027). S.K. also acknowledges the computational support from the Center for Advanced Computation (CAC) at Korea Institute for Advanced Study (KIAS). J.W.R. and S.K were supported by Korea Institute for Advanced Study (KIAS) grant funded by the Korea government.

## Author contributions

J.-W.R and B.A.B. conceived the initial idea and developed the theoretical framework. J.S. contributed to the analysis of theoretical models. S.M., H.L., and S.K. performed the density functional theory calculations. All authors discussed the results and contributed to the writing of the manuscript.

## Competing interests

The authors declare no competing interests.
