## [Transparent Peer Review file · Nature Communications]

Topological Protection by Local Support Symmetry and Destructive Interference

Corresponding Author: Professor Jun-Won Rhim

Version 0:

Reviewer comments:

Reviewer #1

(Remarks to the Author)

The manuscript entitled "Topological Protection by Local Support Symmetry and Destructive Interference" proposes the concept of Local Support Symmetry (LSS) as a means of explaining how symmetry-protected topological features can persist even when the overall Hamiltonian does not possess the corresponding global symmetry. The authors show that if a certain subsystem of a lattice retains a local symmetry, and if the coupling to the symmetry-breaking part satisfies specific algebraic conditions or gives rise to compact localised states through destructive interference, then the topological properties of that subsystem can survive in the full system. They develop a general algebraic framework, support it with several tight-binding examples, and conclude with a density-functional theory study of fluorinated biphenylene, which is presented as a possible material realisation.

In my opinion, the work is clearly written and addresses a problem of wide current interest—the stability of topological features under partial symmetry breaking. The proposed framework bridges different areas of physics: flat-band theory, coupled-layer constructions, and symmetry-protected topology.

However, the novelty of the manuscript lies more in its conceptual synthesis than in a genuinely new physical principle. The central mechanisms that underpin LSS—block decomposition of the Hamiltonian, destructive interference, and the existence of compact localised states—are well established in the literature on flat-band and compact-localisation phenomena. For these reasons, the authors should more clearly acknowledge this connection and present LSS as a unifying and generalisable reformulation rather than as a fundamentally new symmetry class. In my opinion, moderating claims of novelty will make the work stronger and more credible.

It would also help readers if the relationship between LSS and coupled-layer constructions (which are commonly used to build topological crystalline insulators) were clarified.

In fact, the algebraic structure employed here is similar, but the physical intention is reversed: rather than coupling layers to create topology, the LSS framework identifies conditions under which coupling does not destroy it. Framing the paper in this complementary light would situate it naturally among established approaches.

A further weakness of the present version is the absence of a quantitative assessment of robustness. The protection mechanism depends on fine-tuned interference or on special coupling constraints, and its resilience to disorder, longer-range hopping, or structural perturbations remains uncertain. For Nature Communications, it would be important to demonstrate how the predicted topological features evolve under controlled symmetry-breaking perturbations, both in model Hamiltonians and in the DFT-based material. Such analyses would transform the work from a conceptual proposal into an experimentally testable claim.

In conclusion, in its current form, the manuscript offers an interesting and pedagogically useful perspective on a set of known physical ideas. Its main strength is to unify several mechanisms—symmetry protection, destructive interference, and localised-state formation—within a single algebraic and conceptual framework. Yet for publication in Nature Communications, the authors should more carefully situate the work within existing literature, add quantitative tests of robustness, and moderate the claims regarding generality and novelty.

In my opinion, with these improvements, the paper could appeal to a wide readership across condensed-matter theory,

photonics, and materials science. I therefore recommend major revision.

Reviewer #2

(Remarks to the Author)

In this paper, the authors provide a framework for topological protection by local support symmetry (LSS), in which topological protections can persist even when a system respects a symmetry in one part of it while the symmetry is broken in another part. Here, LSS alone cannot preserve these topological characteristics; rather, through matrix analysis they find that the hopping processes between two parts of the system must satisfy a certain condition. In this work, they establish the explicit forms of these conditions for both the insulating and metallic cases, highlighting that a destructive interference of the Bloch wave function can play a crucial role in the fulfillment of these conditions. I think that these results are novel and they expand our understanding of symmetry-protected topological phases.

A minor mistake:

1) On page 8, the 6×6 matrix $h_1(k)$ should be the direct sum of two 3×3 matrices.

Reviewer #3

(Remarks to the Author)

Over the past decade, substantial progress has been made in the study of topological phases. Thanks to intensive efforts, our understanding of phases protected by exact global or crystalline symmetries has become quite mature. In this work, the authors investigate local subspace symmetries (LSS), where a designated subspace (or spatial subregion) respects a symmetry while the rest of the system does not.

I find that the results presented in the manuscript are technically sound. However, in terms of novelty and potential impact on the field, I am afraid that the current manuscript is not sufficiently original or influential to warrant publication in Nature Communications. My assessment is based on the following two points:

(i) Conceptual overlap with "sub-symmetry" (Nature Physics 19, 992-998, 2023).

The paper in Nature Physics introduced sub-symmetry-protected phases, where a symmetry is preserved only within a subspace of the full Hilbert space. For instance, the authors of that work considered a modified SSH chain in which either sublattice A or B preserves chiral symmetry, and they demonstrated that a zero mode remains protected. Much of the present manuscript appears closely related to this framework. While the authors certainly advance the idea, the fundamental concept seems very similar. Could the authors clarify the similarities and distinctions between topological phases protected by LSS and those protected by sub-symmetry?

(ii) Topological features intrinsic to LSS.

As currently presented, LSS-protected phases seem to inherit topology that already exists within the symmetric subspace. If this is the case, it remains unclear whether LSS gives rise to genuinely new phases or topological invariants beyond those already known. Could the authors clarify whether LSS can protect new types of topology that are unique to LSS-protected topological phases?

In addition, I have a few minor comments:

(iii) Scope of protected crossings.

The section "General theory: semimetal case" focuses on crossings enforced by symmetry representations. However, many gapless points are protected by higher-dimensional topological charges (such as a 1D winding number or a Chern number). Could the authors state explicitly whether such topological charges are outside the scope of this work, or explain how (if at all) the LSS framework constrains or incorporates them? A brief discussion of Weyl/Dirac nodes and nodal lines in the presence of LSS would be valuable.

(iv) 3D example.

From a materials perspective, it would be helpful if the authors could present an example involving a three-dimensional system.

Version 1:

Reviewer comments:

Reviewer #1

(Remarks to the Author)

In the new version of their manuscript, the authors have addressed all the points I raised in my previous report. For this reason, I recommend the publication of their work on Nature Communications.

Reviewer #1

Comment 1.

The manuscript entitled "Topological Protection by Local Support Symmetry and Destructive Interference" proposes the concept of Local Support Symmetry (LSS) as a means of explaining how symmetry-protected topological features can persist even when the overall Hamiltonian does not possess the corresponding global symmetry. The authors show that if a certain subsystem of a lattice retains a local symmetry, and if the coupling to the symmetry-breaking part satisfies specific algebraic conditions or gives rise to compact localised states through destructive interference, then the topological properties of that subsystem can survive in the full system. They develop a general algebraic framework, support it with several tight-binding examples, and conclude with a density-functional theory study of fluorinated biphenylene, which is presented as a possible material realisation.

In my opinion, the work is clearly written and addresses a problem of wide current interest—the stability of topological features under partial symmetry breaking. The proposed framework bridges different areas of physics: flat-band theory, coupled-layer constructions, and symmetry-protected topology.

Response.

We sincerely thank the Reviewer for this positive and thoughtful assessment of our work. We are pleased that the Reviewer finds the manuscript clearly written and that the proposed concept of Local Support Symmetry (LSS) and its implications for the robustness of topological features under partial symmetry breaking are viewed as timely and of broad interest. In particular, we appreciate the Reviewer's recognition that our framework connects ideas from flat-band physics, coupled-layer constructions, and symmetry-protected topology, which was one of the central motivations of this study. We have carefully revised the manuscript to further improve clarity and presentation, while preserving the key physical insights highlighted by the Reviewer.

Comment 2.

However, the novelty of the manuscript lies more in its conceptual synthesis than in a genuinely new physical principle. The central mechanisms that underpin LSS—block decomposition of the Hamiltonian, destructive interference, and the existence of compact localised states—are well established in the literature on flat-band and compact-localisation phenomena. For these reasons, the authors should more clearly acknowledge this connection and present LSS as a unifying and generalisable reformulation rather than as a fundamentally new symmetry class. In my opinion, moderating claims of novelty will make the work stronger and more credible.

Response.

We thank the referee for this thoughtful and constructive comment regarding the scope and framing of novelty in our manuscript. After carefully re-examining the manuscript in light of this comment, we found that strong wording suggesting a fundamentally new class of topological phases appeared in two sentences, namely in the Abstract and in the Discussion. Otherwise, throughout the most part of the main text, our intention has been to formulate a general and unifying framework that synthesizes and extends existing mechanisms, such as block decomposition, destructive interference, and compact localization, to a broader setting.

We agree that even these two instances of strong phrasing could give an unintended impression of claiming a new symmetry class or a fundamentally new physical principle. Following the referee's suggestion, we have revised both sentences to moderate the claims and to emphasize more clearly the conceptual and unifying nature of the LSS framework. In the revised version, LSS is presented explicitly as a generalizable formulation for understanding topological protec-

tion in systems with symmetry preserved only on a locally supported subspace, rather than as a new topological classification. We believe that these changes improve the clarity, balance, and credibility of the manuscript, and we sincerely thank the referee for prompting this refinement.

The corresponding revision can be found in Abstract and Discussion:

[Abstract] “We formulate a unifying framework for solid-state phases in which symmetries preserved only within a partial region of the system, termed local support symmetries (LSSs), can protect topological features of the full system, even in the presence of symmetry-breaking couplings.”

[Discussion] “Our theory provides a unifying framework for interpreting protective mechanisms underlying a broad class of topological properties and band crossings in band structures, including situations where the relevant symmetry is preserved only on a restricted subspace.”

Comment 3.

It would also help readers if the relationship between LSS and coupled-layer constructions (which are commonly used to build topological crystalline insulators) were clarified. In fact, the algebraic structure employed here is similar, but the physical intention is reversed: rather than coupling layers to create topology, the LSS framework identifies conditions under which coupling does not destroy it. Framing the paper in this complementary light would situate it naturally among established approaches.

Response.

We thank the referee for this insightful comment and for suggesting a useful conceptual comparison. Let us first clarify what is meant by coupled-layer constructions (CLC) in the context of topological crystalline insulators, and then explain how the present framework relates to, but also differs from, this established approach.

Coupled-layer constructions are a widely used engineering strategy for generating higher-dimensional or crystalline topological phases. In this approach, one starts from lower-dimensional topological building blocks (e.g., 1D SSH chains or 2D Chern/TCI layers) and introduces inter-layer couplings that are constrained by crystalline symmetries. The key idea is that individual layers can already host topologically nontrivial phases, while the specific pattern of symmetry-respecting interlayer couplings organizes and stabilizes these features into a robust higher-dimensional bulk topology with protected boundary states.

The referee is correct that the algebraic structure appearing in our work, namely, a block decomposition of the Hamiltonian and constraints on the inter-block hopping matrices, bears a formal resemblance to those used in CLC. However, the physical intention and logical direction of the two approaches are fundamentally different, as the referee also mentioned. In CLC, inter-layer coupling is deliberately designed to create a new topological phase in the full system. By contrast, the LSS framework starts from a subsystem that already supports a symmetry-protected boundary or band-crossing feature, and asks under what conditions coupling to a symmetry-breaking sector fails to destroy this feature.

In this sense, LSS does not constitute a constructive recipe for generating topology, but rather a stability criterion: it identifies algebraic conditions, often realized through destructive interference or compact localization, under which hybridization with additional degrees of freedom becomes ineffective. Importantly, the protected states in the LSS setting need not originate from stacking or layering, nor from crystalline symmetries acting globally; instead, their robustness follows from the fact that the relevant eigenstates have support only on a symmetry-preserving subspace that is annihilated by symmetry-breaking couplings.

We agree with the referee that framing the present work as complementary to coupled-layer constructions is helpful, and we will revise the discussion to emphasize this point more clearly.

While CLC demonstrates how topology can be engineered by coupling, our work addresses the converse question: when and why topology can survive coupling. We believe that highlighting this distinction better situates the LSS framework within the broader landscape of modern approaches to symmetry-protected and crystalline topological phases.

The corresponding revision can be found in the first paragraph of the discussion section:

“Understanding the role of coupling between block Hamiltonians in shaping topological phases has been a recurring theme in the study of symmetry-protected and crystalline topological matter. A prominent example is the coupled-layer construction, in which couplings between stacked layers give rise to higher-dimensional topological phases. In these approaches, coupling plays a constructive role by generating a global topological obstruction in the full Bloch Hamiltonian. By contrast, the LSS framework focuses on identifying conditions under which coupling fails to destroy pre-existing symmetry-protected features.”

Comment 4.

A further weakness of the present version is the absence of a quantitative assessment of robustness. The protection mechanism depends on fine-tuned interference or on special coupling constraints, and its resilience to disorder, longer-range hopping, or structural perturbations remains uncertain. For Nature Communications, it would be important to demonstrate how the predicted topological features evolve under controlled symmetry-breaking perturbations, both in model Hamiltonians and in the DFT-based material. Such analyses would transform the work from a conceptual proposal into an experimentally testable claim.

Response.

We thank the referee for raising the important issue of quantitative robustness. We fully agree that, for a realistic assessment and experimental relevance, it is essential to characterize how LSS-protected band crossings evolve under controlled symmetry-breaking perturbations.

To address this point, we have introduced a quantitative measure of robustness, which we term the *fragility* of an LSS-protected band crossing. The key observation is that the protection mechanism relies on the stabilization of a compactly supported wave function through destructive interference. When the hopping parameters are modified such that this interference condition is violated, for example, by introducing longer-ranged hopping processes that break the compatibility condition in Eq. (6) in the main text, the band-crossing is generally lifted and a finite gap is induced.

We quantify this effect by defining the fragility parameter

$$F = \Delta/t'_{\max}, \tag{1}$$

where Δ denotes the induced gap and t'_{\max} is the maximum amplitude among the additional (symmetry-breaking) hopping terms. This dimensionless ratio directly measures the sensitivity of the band crossing to perturbations relative to their intrinsic energy scale. In generic situations, one expects $F \sim \mathcal{O}(1)$, indicating that the induced gap is comparable to the perturbative hopping strength. However, the precise value of F depends on how strongly the added hoppings violate the destructive interference condition. In particular, F can vanish in the extreme case where longer-ranged hoppings are present but still satisfy the destructive interference constraints, leaving the band crossing intact. Accordingly, the band crossing becomes increasingly robust as $F \rightarrow 0$.

In the revised manuscript, we have explicitly evaluated the fragility F for all model Hamiltonians considered in the paper, thereby providing a systematic and quantitative assessment of robustness across our examples. Importantly, this framework naturally extends to *ab initio* calculations: longer-ranged hopping amplitudes can be extracted from Wannierized DFT band

structures, allowing the same fragility analysis to be applied to the DFT-based material example discussed in the manuscript. This establishes a direct and quantitative bridge between the conceptual LSS mechanism, realistic microscopic perturbations, and experimentally testable predictions.

The definition of the fragility can be found at the end of the section “General theory: semimetal case”:

“The band crossing protected by the LSS can be lifted when the hopping parameters are tuned such that the compactly supported wave function is no longer stabilized. For instance, this occurs when longer-ranged hopping processes are introduced, thereby breaking the condition in Eq. (6). The resulting gap size depends on the detailed structure of the compactly supported wave function, including its spatial shape and amplitude configuration in real space. We quantify the fragility of the band crossing against such perturbations by the ratio $F = \Delta/t'_{\max}$, where Δ denotes the induced gap and t'_{\max} represents the maximum amplitude among the additional hopping terms. Typically, one expects $F \sim 1$, implying that the induced gap is comparable to the energy scale of the perturbative hopping processes. However, the precise value of F depends on the extent to which these additional hoppings violate the destructive interference condition. In an extreme case, F can vanish if the added hoppings still satisfy the destructive interference condition. In general, the band crossing becomes more robust as F approaches zero.”

In the section “Model-I: topological insulator with local support time-reversal symmetry”, we use the fragility of the band-crossing at the time-reversal invariant momentum of the helical edge states:

“By replacing Δ in the definition of the fragility with the energy splitting of the edge states at $k = 0$, the fragility of the helical edge modes is estimated to be $F \approx 1.57 \times 10^{-5}$.”

In the section “Model-II: Dirac fermions protected by local support C_2 symmetry”, we include the analysis of the fragility and the corresponding figure(Fig. 3) is also modified:

“The condition for destructive interference can be violated by introducing additional hopping processes from the D site to its neighboring A and C sites, as illustrated in the inset of Fig. 3(f). When the hopping amplitudes for these four processes are identical ($\delta t = 0$), the destructive interference remains intact because the amplitudes arriving at the D site from the A and C sites cancel each other. In contrast, when $\delta t \neq 0$, this cancellation no longer occurs and the band crossing becomes gapped. In Fig. 3(f), we quantify the fragility of the band crossing as a function of δt . One finds that the induced band gap remains strongly suppressed for small values of δt , indicating the robustness of the crossing against the longer-range hopping processes.”

In the section “Model-III: Dirac fermions enforced by local support nonsymmorphic symmetry”, we also include the analysis of the fragility and the corresponding figure(Fig. 4) is also modified:

“One can ruin the destructive interference and open a gap by including the hopping processes from A_2 and B_2 sites to the neighboring C_1 and D_1 sites, as illustrated in the inset of Fig. 4(c). The corresponding fragility is plotted as a function of t_4 in Fig. 4(c). One can note that the gap is quite robust for small t_4 .”

In the section “Realistic example: Fluorinated biphenylene network”, we also add the analysis of the fragility:

“The fragilities of the two band crossings, DP_1 and DP_2 , of the unperturbed tight-binding model shown in Fig. 5(d) against longer-ranged hopping processes t_6 , t_7 , and t_8 are evaluated to be $F = 0.00219$ and 0.678 , respectively. The exceptional robustness of DP_1 originates from the fact that the destructive interference of Ψ_1 is approximately preserved even after the inclusion of t_6 , t_7 , and t_8 , since its amplitudes at A_{10} and A_{11} have opposite signs and the magnitudes

of t_7 and t_8 are comparable. In contrast, for Ψ_2 , the amplitudes at A_{10} and A_{11} have the same sign, and as a result, the destructive interference is strongly disrupted.”

Comment 5.

In conclusion, in its current form, the manuscript offers an interesting and pedagogically useful perspective on a set of known physical ideas. Its main strength is to unify several mechanisms—symmetry protection, destructive interference, and localised-state formation—within a single algebraic and conceptual framework. Yet for publication in Nature Communications, the authors should more carefully situate the work within existing literature, add quantitative tests of robustness, and moderate the claims regarding generality and novelty.

In my opinion, with these improvements, the paper could appeal to a wide readership across condensed-matter theory, photonics, and materials science. I therefore recommend major revision.

Response.

We sincerely thank the referee for the careful reading of our manuscript and for the thoughtful overall assessment. We appreciate the referee’s recognition that our main contribution is to unify symmetry protection, destructive interference, and localized-state formation within a single algebraic and conceptual framework, which can serve a pedagogical role and help clarify the common structure behind these seemingly distinct mechanisms. We also take seriously all the referee’s concerns regarding positioning within the literature, quantitative robustness, and the tone of our claims. We believe these changes substantially strengthen the manuscript and clarify its scope, while preserving the conceptual message highlighted by the referee. We are grateful for the recommendation for major revision, and we hope that the revised version now meets the standards for publication in *Nature Communications* and will indeed be of interest to readers across condensed-matter theory, photonics, and materials science.

Reviewer #2

Comment 1.

In this paper, the authors provide a framework for topological protection by local support symmetry (LSS), in which topological protections can persist even when a system respects a symmetry in one part of it while the symmetry is broken in another part. Here, LSS alone cannot preserve these topological characteristics; rather, through matrix analysis they find that the hopping processes between two parts of the system must satisfy a certain condition. In this work, they establish the explicit forms of these conditions for both the insulating and metallic cases, highlighting that a destructive interference of the Bloch wave function can play a crucial role in the fulfillment of these conditions. I think that these results are novel and they expand our understanding of symmetry-protected topological phases.

Response.

We thank the referee for this positive and accurate summary of our work. We are pleased that the referee recognizes the novelty of the proposed framework and its contribution to broadening the understanding of symmetry-protected topological phases.

Comment 2.

A minor mistake: (1) On page 8, the 6×6 matrix $h_1(k)$ should be the direct sum of two 3×3 matrices.

Response.

We thank the referee for pointing out this mistake. We have corrected the corresponding part

The corresponding revision can be found on Page X, Paragraph Y:

“The \mathcal{S}_1 -part block of the Hamiltonian (??) is given by a 6×6 matrix $h_1(\mathbf{k}) = h_{1,\uparrow}(\mathbf{k}) \oplus h_{1,\downarrow}(\mathbf{k})$, where...”

Reviewer #3

Comment 1.

Over the past decade, substantial progress has been made in the study of topological phases. Thanks to intensive efforts, our understanding of phases protected by exact global or crystalline symmetries (LSS), where a designated subspace (or spatial subregion) respects a symmetry while the rest of the system does not.

I find that the results presented in the manuscript are technically sound. However, in terms of novelty and potential impact on the field, I am afraid that the current manuscript is not sufficiently original or influential to warrant publication in Nature Communications. My assessment is based on the following two points:

Response.

We thank the referee for the careful reading of our manuscript and for the positive assessment of the technical correctness of our results. We also appreciate the referee's comments regarding the novelty and potential impact of the work, which have helped us to clarify and strengthen the central contributions of the manuscript. In the revised manuscript, we have substantially improved the presentation to more clearly articulate the novelty and significance of our results. We now respond in detail to the two specific points raised by the referee below.

Comment 2.

Conceptual overlap with "sub-symmetry" (Nature Physics 19, 992-998, 2023). The paper in Nature Physics introduced sub-symmetry-protected phases, where a symmetry is preserved only within a subspace of the full Hilbert space. For instance, the authors of that work considered a modified SSH chain in which either sublattice A or B preserves chiral symmetry, and they demonstrated that a zero mode remains protected. Much of the present manuscript appears closely related to this framework. While the authors certainly advance the idea, the fundamental concept seems very similar. Could the authors clarify the similarities and distinctions between topological phases protected by LSS and those protected by sub-symmetry?

Response.

We thank the referee for pointing out the conceptual relation to the recent work on sub-symmetry protection (Nature Physics **19**, 992–998 (2023)). Below we briefly summarize that work and then clarify the essential differences from the present manuscript.

Summary of the sub-symmetry approach. The Nature Physics paper introduced the concept of *sub-symmetry* (SubSy), in which a protecting symmetry is preserved only within a restricted subspace of the full Hilbert space, typically associated with a single sublattice. Using the SSH chain and the breathing kagome lattice as prototypical examples, the authors demonstrated that even when the global protecting symmetry and the corresponding bulk topological invariant are destroyed, certain boundary zero modes can remain pinned at zero energy if the perturbations respect the SubSy. In this framework, the protection mechanism relies on the fact that the boundary zero mode is entirely supported on a specific subspace (for example, a single sublattice). SubSy-preserving perturbations are defined such that they do not couple to this subspace, or act trivially on it, which implies that the perturbation operator annihilates the boundary-state wave function. As a consequence, although the perturbations generally break the global protecting symmetry and destroy the bulk topological invariant, the boundary state remains an exact eigenstate with its eigenvalue pinned at zero energy.

Key distinctions from the present work. While both approaches recognize that symmetry acting on a restricted subspace can have physical consequences, the scope, mechanisms, and implications are fundamentally different. The SubSy framework addresses the protection

of *specific boundary zero modes* after the bulk topology has been destroyed. By contrast, the Local Support Symmetry (LSS) framework developed in the present manuscript provides a general algebraic criterion on the Hamiltonian that ensures the protection of *topological features themselves*, including band crossings and topological invariants, and applies to both insulating and metallic systems. In our work, the protection originates from destructive interference and the resulting local support of eigenstates, rather than from projector-based annihilation of perturbations on a single sublattice.

In this sense, the conceptual overlap is limited to the general observation that a symmetry need not act on the full Hilbert space to be physically relevant, whereas the physical objects being protected, the underlying mechanisms, and the range of applicability are completely distinct.

The corresponding revision can be found at the final paragraph in the discussion section:

“We note a conceptual relation to the recently introduced notion of sub-symmetry protection, where a symmetry preserved only within a restricted subspace can protect certain boundary zero modes even after the bulk topological invariant is destroyed. In that case, protection arises because the boundary state is fully supported on the subspace on which the sub-symmetry acts, so that symmetry-preserving perturbations do not shift its eigenvalue. By contrast, the LSS framework developed here protects topological features themselves through algebraic constraints on the Hamiltonian and the destructive interference plays a essential role.”

Comment 3.

Topological features intrinsic to LSS. As currently presented, LSS-protected phases seem to inherit topology that already exists within the symmetric subspace. If this is the case, it remains unclear whether LSS gives rise to genuinely new phases or topological invariants beyond those already known. Could the authors clarify whether LSS can protect new types of topology that are unique to LSS-protected topological phases?

Response.

We thank the referee for raising this important conceptual question regarding whether LSS gives rise to genuinely new topological phases or invariants. As pointed out by the referee, we agree that the topological features protected by LSS, as presented in this work, are inherited from the symmetry-preserving subspace and can be characterized using conventional topological invariants already established in the literature. In this sense, LSS does not introduce a new topological classification or fundamentally new invariants.

At the same time, following the another referee’s suggestion to emphasize the conceptual and unifying nature of the LSS framework, we have carefully revised the manuscript to clarify the intended scope of our contribution. The central aim of LSS is not to generate new topology, but to address a complementary and previously underexplored question: under what conditions do topological features defined within a restricted, symmetry-preserving subspace remain physically robust when the corresponding symmetry is broken in the full Hamiltonian.

From this perspective, LSS provides a general and physically transparent framework for understanding the stability of known topological properties and band crossings under partial symmetry breaking. While the underlying invariants themselves are conventional, LSS identifies broad and model-independent conditions, such as block structure and destructive interference, under which these invariants continue to govern observable boundary or band-crossing phenomena in systems that would otherwise be classified as topologically trivial within standard symmetry-based approaches. We have clarified this point in the revised manuscript and believe that this framing better reflects the conceptual contribution and intended scope of the work.

The corresponding revision can be found in Abstract and Discussion:

[Abstract] “We formulate a unifying framework for solid-state phases in which symmetries preserved only within a partial region of the system, termed local support symmetries (LSSs),

can protect topological features of the full system, even in the presence of symmetry-breaking couplings.”

[Discussion] “Our theory provides a unifying framework for interpreting protective mechanisms underlying a broad class of topological properties and band crossings in band structures, including situations where the relevant symmetry is preserved only on a restricted subspace.”

Comment 4.

Scope of protected crossings. The section “General theory: semimetal case” focuses on crossings enforced by symmetry representations. However, many gapless points are protected by higher-dimensional topological charges (such as a 1D winding number or a Chern number). Could the authors state explicitly whether such topological charges are outside the scope of this work, or explain how (if at all) the LSS framework constrains or incorporates them? A brief discussion of Weyl/Dirac nodes and nodal lines in the presence of LSS would be valuable.

Response.

We thank the referee for raising this important question regarding the scope of protected crossings and the role of higher-dimensional topological charges such as winding numbers, Chern numbers, and chiral charges.

Our framework indeed focuses on band crossings enforced by local-support symmetries (LSSs) through representation constraints of the inter-part Hamiltonian. In this sense, the “General theory: semimetal case” addresses symmetry-enforced gapless points rather than topologically protected ones. We now clarify this explicitly in the revised manuscript.

(i) Dirac and nodal-line semimetals: For Dirac points and nodal lines, crystalline symmetries, such as inversion, mirror, and rotational symmetries, play a central role in protecting the gapless structures. Within our framework, one can indeed construct tight-binding models in which such crystalline symmetries act locally on one part of the system and are effectively realized through the LSS constraints, even when the global symmetry of the full system is broken. In other words, as long as the inter-part coupling satisfies the condition for the inter-part coupling derived in the section “General theory: semimetal case,” the Bloch wave functions localized in the symmetric part remain compactly supported and retain the Dirac or nodal-line structure. A nodal line semimetal example is discussed in the next question.

(ii) Weyl semimetals: The situation for Weyl nodes is fundamentally different. Their stability does not rely on crystalline symmetries but rather on their topological monopole charge; thus, Weyl points do not require symmetry protection. For this reason, the analysis of Weyl nodes is not the principal target of the LSS framework, which is designed to capture symmetry-enforced, rather than topologically enforced, semimetallic structures.

Nonetheless, the LSS point of view can still offer insight. One can design models in which the wave functions forming the Weyl cones are compactly localized to a subsystem through destructive interference, provided the inter-part hopping satisfies the LSS compatibility condition. This ensures that the Weyl-cone sector is preserved within one part of the lattice. One may also analyze, within the same framework, which forms of inter-part coupling prevent two Weyl nodes with opposite chiralities from merging and annihilating.

We have added a brief discussion in the discussion section in the revised text as follows.:

“While we have examined this mechanism primarily in two-dimensional examples, it is equally applicable to three-dimensional systems, including Dirac and nodal-line semimetals. Dirac points and nodal lines, typically protected by crystalline symmetries such as inversion, mirror, or rotation, can still appear within our framework when these symmetries act locally on the \mathcal{S}_1 subsystem and the inter-part coupling satisfies the condition in Eq. (6). As a concrete illustration, Supplementary Sec. V demonstrates that a nodal line in a three-dimensional model can be protected by a local support mirror symmetry. In contrast, Weyl nodes fall outside the

primary scope of LSS protection, as their stability originates from a topological monopole charge rather than a crystalline symmetry. Nevertheless, if the inter-part hopping respects the LSS compatibility condition, destructive interference may still confine the Weyl-cone wave functions to \mathcal{S}_1 , thereby constraining how opposite-chirality Weyl nodes hybridize or annihilate through couplings between the \mathcal{S}_1 and \mathcal{S}_2 parts.”

Comment 5.

3D example. From a materials perspective, it would be helpful if the authors could present an example involving a three-dimensional system.

Response.

We thank the referee for a fruitful suggestion. As a 3D example, we constructed a simple nodal line semimetal model, whose ring-shape band-crossing is protected by a local support mirror symmetry. We mentioned it in the discussion section of the revised manuscript as:

“As a concrete illustration, Supplementary Sec. V demonstrates that a nodal line in a three-dimensional model can be protected by a local support mirror symmetry.

Details are included In Supplementary Information as:

“We propose a three-dimensional toy model that hosts a ring-shaped nodal line protected by a local-support mirror symmetry along the z -axis. We start from a subsystem, denoted by \mathcal{S}_1 , which possesses a global mirror symmetry with respect to the z -axis, as illustrated in Fig. 5(a). This subsystem is constructed on a cubic lattice containing two sites per unit cell. We assume that s -orbitals reside on these sites. The sub-Hamiltonian for \mathcal{S}_1 , consisting of A and B sites, is given by

$$h_1(\mathbf{k}) = \begin{pmatrix} 0 & f(\mathbf{k}) \\ f(\mathbf{k})^* & 0 \end{pmatrix}, \quad (2)$$

where

$$f(\mathbf{k}) = -t_x(1 + e^{ik_z}) \cos k_x - t_y(1 + e^{ik_z}) \cos k_y - t_z(1 + e^{ik_z}) - it_1(1 - e^{ik_z}). \quad (3)$$

We have a nodal line at $k_z = 0$ plane for $t_x = t_y = t_1 = 1$ and $t_z = -0.2$. This band-crossing is protected by the mirror symmetry with respect to the z -axis, whose mirror plane is indicated in Fig. 5(a). The mirror operation is represented by $M_z = \sigma_x$.

We now introduce two additional sites, labeled C and D, as shown on the right-hand side of Fig. 5(a). The subsystem composed of these two sites is denoted as \mathcal{S}_2 . When \mathcal{S}_1 and \mathcal{S}_2 are coupled, the overall system no longer preserves the mirror symmetry. We assume that the C-site hosts an s -orbital, while the D-site is occupied by a p_z -orbital. Consequently, the hopping between the neighboring B and C sites vanishes, and the hopping amplitudes from the D-site to the adjacent upper and lower A-sites acquire opposite signs due to the odd parity of the p_z orbital. The sub-Hamiltonian for the \mathcal{S}_2 part is given by

$$h_2(\mathbf{k}) = \begin{pmatrix} \epsilon_0 & \alpha + \beta e^{-ik_z} \\ \alpha + \beta e^{ik_z} & \epsilon_0 \end{pmatrix}, \quad (4)$$

where we set $\epsilon_0 = -2$, $\alpha = 1$, and $\beta = 0.5$. On the other hand, the inter-part Hamiltonian is given by

$$h_{12}(\mathbf{k}) = \begin{pmatrix} u & 0 \\ u & w(1 - e^{ik_z}) \end{pmatrix}, \quad (5)$$

where the second column reflects the properties of hopping processes between the p_z orbital at the D site and s orbitals at A and B sites, which are mentioned above. According to the general

theory for semimetal cases in the maintext, the columns of the inter-part Hamiltonian should be proportional to the eigenstate of the symmetry operation with an eigenvalue 1 or vanish at symmetry-invariant momenta for the nodal line to be protected although mirror symmetry is broken. In this nodal line semimetal model, the inter-part coupling indeed fulfills these conditions. The first column of $h_{12}(\mathbf{k})$ is proportional to $(1, 1)^T$, which is the eigenvector of M_z with an unity eigenvalue, and the second column vanishes at $k_z = 0$. Consequently, the nodal line between the two bands of $h_1(\mathbf{k})$ at $k_z = 0$ remains intact even when the parameters u and w are switched on, as shown in Fig. 5(b). The two crossing bands carry opposite local-support mirror eigenvalues, and the eigenstates with mirror eigenvalue -1 exhibit compact localization, having vanishing amplitudes on the C and D sites. Such eigenstates are stabilized through destructive interference: the opposite amplitudes on the A and B sites cancel at the C site, ensuring that the wave function remains confined within \mathcal{S}_1 . To quantify the robustness of the nodal line, we compute its fragility, defined as $\Delta/\delta t$, against an additional hopping term that breaks the local-support condition by modifying the (2, 2) element of $h_{12}(\mathbf{k})$ to $w(1 - e^{ik_z}) + \delta t$. Here, Δ denotes the maximum gap opened along the nodal line. As shown in Fig. 5(c), the fragility decreases as δt increases, since the induced gap grows sublinearly as a function of δt .

Reviewer #1 (Remarks to the Author): In the new version of their manuscript, the authors have addressed all the points I raised in my previous report. For this reason, I recommend the publication of their work on Nature Communications.

We thank the reviewer for the positive assessment of our manuscript.

We are pleased to hear that all points raised in the previous report have been satisfactorily addressed and appreciate the reviewer's recommendation for publication in *Nature Communications*.

No further changes were required in response to this comment.